# Discovery of processive catalysis by an exo-hydrolase with a pocket-shaped active site

Victor A. Streltsov[1,15], Sukanya Luang[2,15], Alys Peisley [1], Joseph N. Varghese[1], James R. Ketudat Cairns [3], Sebastien Fort [4], Marcel Hijnen[5], Igor Tvaroška[6], Ana Ardá[7], Jesús Jiménez-Barbero[7], Mercedes Alfonso-Prieto [8], Carme Rovira [8,9], Fernanda Mendoza [10,11], Laura Tiessler-Sala[11], José-Emilio Sánchez-Aparicio [11], Jaime Rodríguez-Guerra [11,12], José M. Lluch [11,13], Jean-Didier Maréchal[11], Laura Masgrau [11,13] & Maria Hrmova [2,14]

Substrates associate and products dissociate from enzyme catalytic sites rapidly, which hampers investigations of their trajectories. The high-resolution structure of the native *Hordeum* exo-hydrolase HvExoI isolated from seedlings reveals that non-covalently trapped glucose forms a stable enzyme-product complex. Here, we report that the alkyl β-D-glucoside and methyl 6-thio-β-gentiobioside substrate analogues perfused in crystalline HvExoI bind across the catalytic site after they displace glucose, while methyl 2-thio-β-sophoroside attaches nearby. Structural analyses and multi-scale molecular modelling of nanoscale reactant movements in HvExoI reveal that upon productive binding of incoming substrates, the glucose product modifies its binding patterns and evokes the formation of a transient lateral cavity, which serves as a conduit for glucose departure to allow for the next catalytic round. This path enables substrate-product assisted processive catalysis through multiple hydrolytic events without HvExoI losing contact with oligo- or polymeric substrates. We anticipate that such enzyme plasticity could be prevalent among exo-hydrolases.

[1] Commonwealth Scientific and Industrial Research Organisation, Materials Science and Engineering, Parkville Victoria 3052, Australia. [2] School of Agriculture, Food and Wine, University of Adelaide, Waite Campus, Glen Osmond South Australia 5064, Australia. [3] School of Chemistry and Center for Biomolecular Structure, Function and Application, Suranaree University of Technology, Nakhon Ratchasima 30000, Thailand. [4] University Grenoble Alpes, Centre de Recherches sur les Macromolécules Végétales, Grenoble cedex 9 38041, France. [5] GE Healthcare Life Sciences, Paramatta NSW 2150, Australia. [6] Institute of Chemistry, Slovak Academy of Sciences, Bratislava 84538, Slovak Republic. [7] Centre for Cooperative Research in Biosciences, Derio-Bizkaia 48160, Spain. [8] Departament de Química Inorgànica i Orgànica, Universitat de Barcelona, Barcelona 08028, Spain. [9] Institució Catalana de Recerca i Estudis Avançats, Barcelona 08010, Spain. [10] Departamento de Ciencias Químicas, Universidad Andrés Bello, Sede Concepción, Talcahuano 4260000, Chile. [11] Departament de Química, Universitat Autònoma de Barcelona, Bellaterra 08193, Spain. [12] Institute of Chemical Research of Catalonia, The Barcelona Institute of Science and Technology, Tarragona 43007, Spain. [13] Institut de Biotecnologia i de Biomedicina, Universitat Autònoma de Barcelona, Bellaterra 08193, Spain. [14] School of Life Sciences, Huaiyin Normal University, Huaian 223300, China. [15] These authors contributed equally: Victor A. Streltsov and Sukanya Luang. Correspondence and requests for materials should be addressed to M.H. (email: maria.hrmova@adelaide.edu.au)

Enzymes are biological catalysts that are fundamental to life. Enzymes afford enormous accelerations to chemical reactions compared to uncatalysed reactions. It has become increasingly recognised that the largest contribution to the enzyme catalytic power arises from the electrostatic environment of polar active sites[1,2]. Enzymes use protein architectures to precisely position a set of amino acid residues to catalyse inter-conversions of substrates into products. This structure-based view is supported by thermodynamic and kinetic models that treat enzymes in complex with substrates and transition states in succession[3]. Only rarely these models consider substrate and product diffusion as a part of their catalytic mechanisms, as these processes proceed rapidly, although on occasions products are seen entrapped in active sites[4]. Yet, the lack of descriptions of substrate associative and product dissociative pathways creates a knowledge gap[5].

A vital conclusion drawn from the structural studies with the native β-D-glucan glucohydrolase, isoenzyme ExoI (HvExoI) isolated from barley seedlings, was that the glucose (Glc) product released from β-D-glucoside substrates remains entrapped in the enzyme active site until an incoming substrate binds[6–9], presumably lowering the energy barrier to facilitate Glc displacement. At this stage, the mechanism of Glc displacement and how it is linked to the catalytic cycle of HvExoI remained unanswered. Notably, no other native GH3 structures with entrapped Glc are available, because these recombinant enzymes not exposed to presumably high enough Glc concentration, do not entrap Glc during protein maturation and secretion. However, several GH3 enzyme complexes with in crystallo-perfused Glc[10,11] or Glc-derivatives[12,13] are available. Additionally, the native GH78 α-L-rhamnosidase with a deep pocket-shaped active site also holds the entrapped Glc molecule, which was not perfused in crystals[14].

We have previously observed that mechanism-based inhibitors conduritol B epoxide and 2′,4′-dinitrophenyl 2-deoxy-2-fluoro-β-D-glucoside dislodge Glc from the −1 subsite and form stable cyclitol esters via an α-anomeric linkage with Asp285[7]. Similarly, transition-state ion-like gluco-phenylimidazole mimics displace Glc and bind in the −1 subsite with their Glc component distorted in the $^4E$ envelope conformation[15,16]. Non-hydrolysable S-glycoside analogues also remove Glc from the −1 subsite and bind across the active site[7,17]. Because HvExoI hydrolyses various positional isomers of β-D-glucosides, we rationalised this obser-vation by the non-reducing Glc moiety being locked into a fixed position via hydrogen bonds (H-bonds) with the residues in the −1 subsite, whereas the reducing-end moiety is free to adopt multiple orientations in the +1 subsite. This allows the remainder of the saccharide substrates beyond the +1 subsite to project unencumbered[7,9,17].

The fundamental function of hydrolysis of oligo- and poly-saccharides is essential for the understanding of the global carbon cycle, which forms the basis of a multibillion-dollar biotechno-logical industry. In this context, the HvExoI enzyme[6] that belongs to the glycoside hydrolase (GH) family 3[18], serves as one of the archetypal models for hydrolysis of oligo- and polysaccharides. Structural information of GH3 enzymes is available through 36 unique entries in the Protein Data Bank (PDB) from all phyla, although there is only one plant structure available (that of HvExoI), notably in complex with a variety of oligosaccharide substrate analogues and mimics[7–9]. HvExoI folds in an (α/β)$_8$ barrel (domain 1) and an (α/β)$_6$ sandwich (domain 2) that at their interface cradle a 13 Å-deep pocket- or crater-shaped active site. This site accommodates the catalytic nucleophile Asp285 and catalytic acid/base Glu491 in the −1 subsite, while the aromatic clamp of Trp286 and Trp434 indole moieties constitutes the neighbouring +1 subsite[6–9]. HvExoI releases a single Glc from

non-reducing termini of oligo- and polymeric β-D-glucosides to support seed germination and plant development[9].

In this work, we examine product and substrate pathways along the catalytic cycle of a plant exo-hydrolase HvExoI, and how this enzyme utilises its plasticity and that of glycoside substrates. We use the high-resolution X-ray crystallography supported with enzyme kinetics, mass spectrometry, nuclear magnetic resonance (NMR) spectroscopy, multi-scale molecular modelling employing docking[19], Molecular dynamics (MD) simulations, Genetic Algorithms with Unrestricted Descriptors for Intuitive Molecular Modelling (GaudiMM)[20] and Protein Energy Landscape Exploration (PELE)[21] calculations, to reveal the Glc product displacement route, and to extract how each hydrolytic event including Glc release is precisely coordinated with the incoming substrate association and hydrolysis. This leads us to generate a variant enzyme and probe it for Glc entrapment and the S-glycoside analogue binding. Along this journey with HvExoI, we observe a remarkable phenomenon, coin the term 'substrate-product assisted processive catalysis', and describe how product and substrate trajectories are coordinated via this mechanism during catalysis. We discover that through chemical signalling, the incoming substrate evokes the formation of an autonomous and transient lateral cavity that serves as a conduit for the Glc product displacement from the active site. Crucially, these co-operative reactant pathways, where polymeric substrates never dissociate from the enzyme, sustain consecutive relocations of substrates and products via dedicated routes, ensuring that this exo-hydrolase adopts processive catalysis.

## Results

**GC/MS proves that Glc is bound to crystalline native HvExoI.** We purified[22], crystallised[23] and chemically analysed crystals to reconcile our structural observations with Glc entrapment in native HvExoI purified from barley seedlings[6–9]. Several crystals removed from the mother liquor were extracted and fractionated by normal-phase high performance liquid chromatography (HPLC) with evaporative light scattering detection (Fig. 1a; top panel, solid line). The mother liquor from these crystals was also analysed, both without and augmented with authentic Glc (Fig. 1a; top panel, dashed and dotted lines, respectively). The materials from these preparations eluting between 6.0 and 7.3 min during HPLC separation were pooled, reduced and per-acetylated. The total ion chromatograms of the material extracted from crystals and from the mother liquor augmented with Glc showed a peak with the retention time of 23.26 min, corre-sponding to glucitol hexa-acetate (Fig. 1b; 1st and 3rd middle panels). Mass fragmentation spectra of these materials (Fig. 1c; bottom panels) showed identical profiles of total ions, demon-strating that the chemical substance eluting at 23.26 min was Glc. We could not detect Glc in the mother liquor, from which crystals had been removed (Fig. 1b; middle panel), while authentic Glc added to the same mother liquor was readily identifiable.

**Crystal structures reveal the Glc product in native HvExoI.** The key observation that led us to explore the precise spatial disposition of the Glc product in the active site and the Glc displacement route in HvExoI, was based on our original obser-vation that the Glc product remains entrapped in the enzyme's active site that is isolated from the native source of young seedlings[6,8]. We anticipated that the last remaining Glc molecule originating from oligo- and polysaccharide substrates, stays associated with the enzyme in plant tissues. To this end, previous crystal structures of native HvExoI at 2.20 Å and 2.70 Å (in-house X-ray source data) resolution showed a single Glc molecule bound in the −1 subsite[6,8]. In the current work we refined native

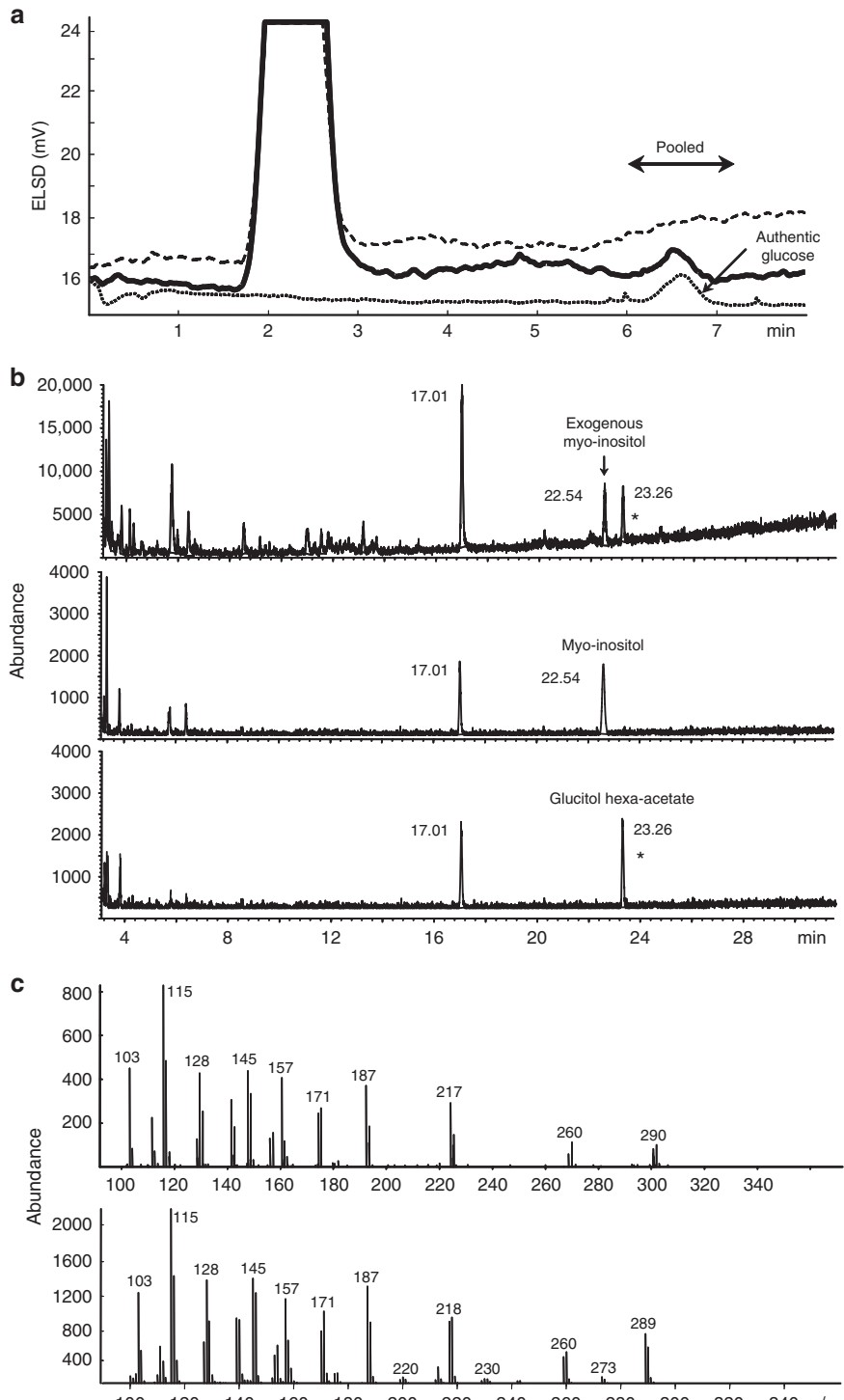

**Fig. 1** Glc trapped in native HvExoI is bound non-covalently in the $^4C_1$ conformation. **a** HPLC chromatogram of material extracted from crystals (solid line), mother liquor (dashed) and mother liquor augmented with authentic Glc (dotted). Arrow points to the peak of authentic Glc in augmented mother liquor. Fractions (left right arrow) were pooled for GC/MS analysis. **b** Total ion chromatograms obtained by GC/MS of HPLC-eluted materials (6–7.3 min) containing alditol acetates. Materials from crystals (top), mother liquor (middle) and mother liquor augmented with Glc (bottom). Myo-inositol (20 ng) served as an internal standard (top and middle panels). Numbers near peaks indicate retention times. Substance eluted in 17.01 min peak corresponds to a contaminating plasticiser. Material extracted from crystals and eluting at 23.26 min (top panel) is identical to that of glucitol hexa-acetate at 23.26 min (bottom panel) on a Prevail Carbohydrate ES column (peaks indicated by asterisks). **c** Fragmentation mass spectra of glucitol hexa-acetate: top, material extracted from crystals; bottom glucitol hexa-acetate eluting at 23.26 min

HvExoI at 1.65 Å resolution (synchrotron data), revealing that Glc is bound at 0.5 occupancy at each −1 and +1 subsites in the $^4C_1$ chair conformation, suggesting that it may be mobile between both subsites (Fig. 2a; Supplementary Fig. 1a).

**Glc is absent in recombinant HvExoI but could be perfused in.** Although Glc has always been detected in the native enzyme isolated from barley seedlings[6,8], it was uncertain if it could be observed in recombinant HvExoI produced in *Pichia pastoris*. The 1.45 Å structure of the recombinant enzyme, which is kinetically and structurally nearly identical to the native one[24,25], failed to show the presence of Glc in the active site, where we identified glycerol and up to seven water molecules mimicking the positions of hydroxyl groups of Glc bound in native HvExoI (Fig. 3a; Supplementary Fig. 2a). However, after the crystals of recombinant HvExoI were perfused with Glc at a near saturating concentration, the 1.55 Å structure revealed two Glc molecules one each in the −1 and +1 subsites, adopting alternate $^4C_1$ chair (occupancy 0.8) and $^1S_3$ skew boat (occupancy 0.2) conformations with classical Cremer–Pople ring-puckering parameters[26]. In the −1 subsite, we detected the network of 12–13 mono- and co-operative bi- and tri-dentate H-bonds[27,28]; of interest were those forming short H-bonds of 2.5–2.7 Å between Oδ1 and Oδ2 of Asp95, and the C6-OH and C4-OH groups of the Glc moiety. We attempted to position into the electron density other skew boat conformers of Glc ($^3S_5$, $^1S_5$) with various occupancies, however, convergence in the refinement was reached with those of $^4C_1$ and $^1S_3$. We concluded that during maturation in *P. pastoris*, Glc cannot be entrapped and thus observed, because the intracellular concentration in this host is not high enough during enzyme maturation or secretion. To demonstrate that Glc was not displaced from the active site by the cryoprotectant, data were collected from recombinant crystals at ambient temperatures without applying vitrification solutions. Here, no Glc was observed in the active site, but was readily observable after perfusion of crystals with Glc.

**SPR analysis of Glc and substrate binding to HvExoI.** To determine the strength of Glc binding and to compare it with that of the substrate thio-analogue methyl 6-thio-β-gentiobioside (G6SG-OMe), we used the recombinant HvExoI enzyme. The steady-state affinity[29] $K_D$ value of $0.008 \times 10^{-3}$ M for G6SG-OMe revealed that it bound tighter than the Glc hydrolytic product ($K_D$ of $0.16 \times 10^{-3}$ M) (Table 1; Supplementary Fig. 3).

**NMR spectroscopy of Glc binding to recombinant HvExoI.** To identify conformational states of bound Glc with the measured $K_D$ value of $0.16 \times 10^{-3}$ M (Table 1; Supplementary Fig. 3), we used $^1$H saturation transferred difference (STD)[30] and transferred nuclear Overhauser effect spectroscopy (trNOESY)[31] in solution NMR. STD spectra recorded with Glc in a 60-fold excess relative to HvExoI detected Glc bound non-covalently. As only STDs for protons of β-anomeric Glc (and not for α-form) were observed, while one would expect an α:β anomeric ratio of ~2:3 in solution equilibrium, we concluded that the enzyme specifically binds β-anomeric Glc (Fig. 4a); this observation supports the retaining hydrolytic mechanism of HvExoI[22]. To identify the conformation of Glc, trNOESY experiments under a relatively low HvExoI versus Glc ratio revealed weak trNOEs between H1-H3 and H1-H5 protons of β-anomeric Glc (Fig. 4b), consistent with β-anomeric Glc bound in the low-energy $^4C_1$ chair conformation. To corroborate these findings, thiocellobiose which cannot be hydrolysed by HvExoI[7], was added in excess relative to HvExoI. Here, clear negative NOEs between specific thiocellobiose proton pairs were observed (Fig. 4c), while strong trNOEs of the

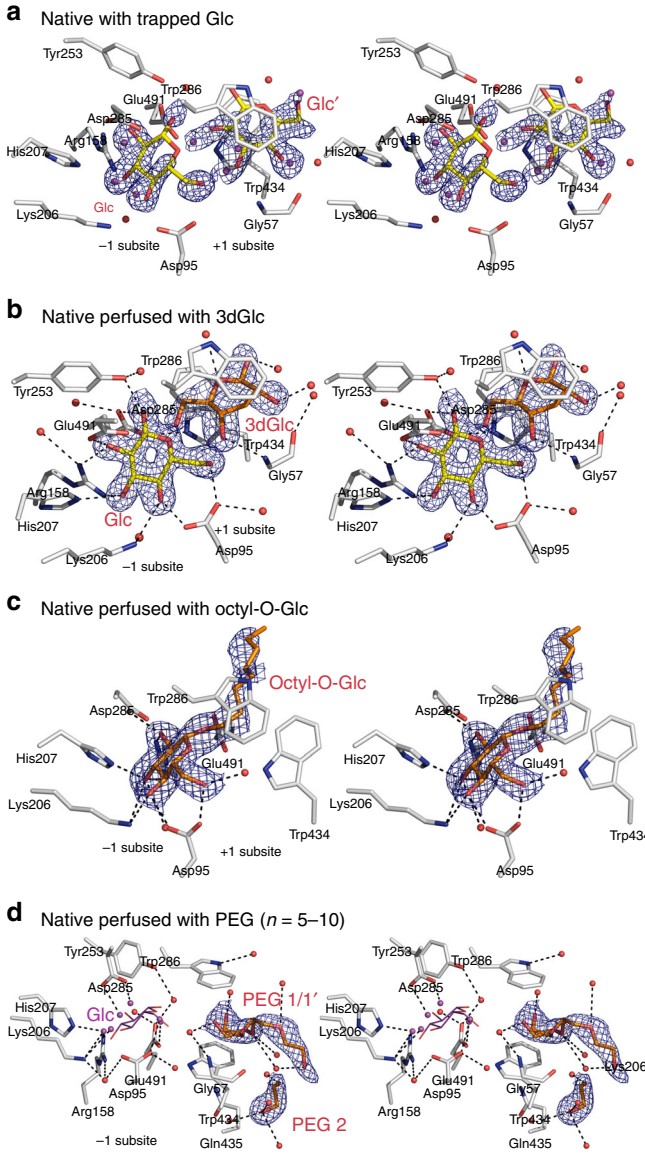

**a** Native with trapped Glc

**b** Native perfused with 3dGlc

**c** Native perfused with octyl-O-Glc

**d** Native perfused with PEG (*n* = 5–10)

**Fig. 2** Native HvExoI with trapped Glc, and with perfused 3dGlc, octyl-O-Glc and PEG. **a** Stereo view of native HvExoI with Glc (Glc and Glc′, carbons: yellow sticks) at 0.5 occupancy, which oscillates between the −1 and +1 subsites. **b** Stereo view of native HvExoI with Glc (carbons: yellow sticks) and 3dGlc (carbons: orange sticks) at 1.0 occupancies, bound in the −1 and +1 subsites, respectively. **c** Stereo view of native HvExoI with octyl-O-Glc (carbons: orange sticks) at 1.0 occupancy, bound across the −1 and +1 subsites. **d** Stereo view of native HvExoI with two PEG (*n* = 5–10) molecules (carbons: orange sticks; PEG 1 in two alternate conformations at occupancies 0.5 each and PEG 2 at occupancy 1.0), bound in the +1 and putative +2 subsites. Grey, red, and blue represent carbon, oxygen and nitrogen atoms, respectively. Water molecules are shown as red or magenta (bound alternate water molecules when the ligand is missing and are not numbered) spheres in complexes with Glc and PEG. In the PEG complex magenta-coloured water molecules mimic positions of OH groups of Glc (added for illustration in magenta lines). Separations of less than 3.50 Å from active site residues (carbons: grey sticks) are shown as dashed lines. Derived |2mFobs − DFcalc| electron density maps are contoured at 1σ (blue mesh). In |2mFobs − DFcalc|, Fobs and Fcalc are observed and calculated X-ray structure factor amplitudes, where m is the figure of merit and D is the estimated coordinate error

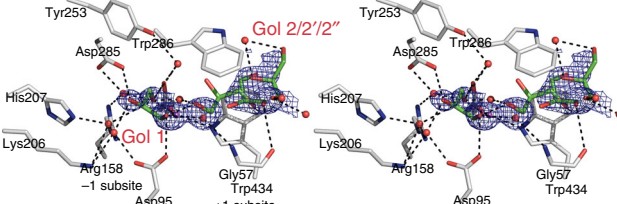

**a** Recombinant in ligand-free form

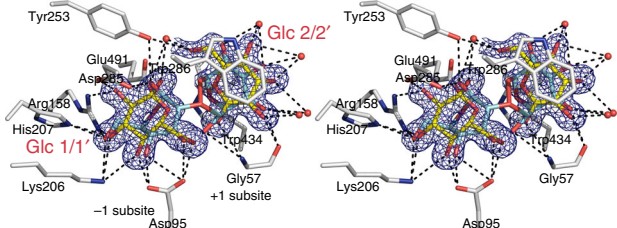

**b** Recombinant perfused with Glc

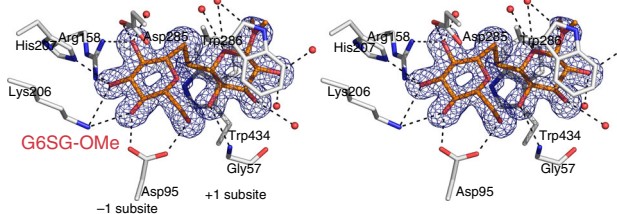

**c** Recombinant perfused with G6SG-OMe

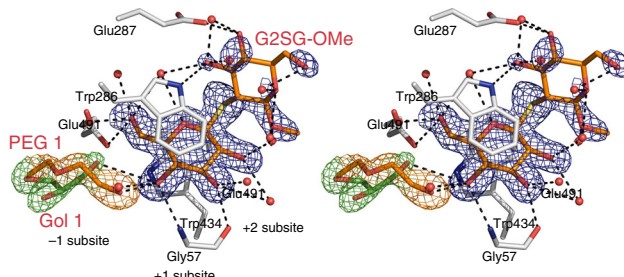

**d** Recombinant perfused with G2SG-OMe

**Fig. 3** Recombinant HvExoI and with perfused Glc, G6SG-OMe, or G2SG-OMe. **a** Stereo view of recombinant HvExoI in a ligand-free form. Two glycerol molecules (carbons: green sticks) (Gol 1 at occupancy 0.5; Gol 2 in three alternate conformations) are bound in the −1 and +1 subsites. **b** Stereo view of recombinant HvExoI with two Glc molecules in the $^4C_1$ conformation (carbons: yellow sticks) at 0.8 occupancy and in the $^1S_3$ conformation (carbons: cyan sticks) at 0.2 occupancy, bound in the −1 and +1 subsites. **c** Stereo view of recombinant HvExoI with G6SG-OMe (carbons: orange sticks) at 1.0 occupancy, bound across the −1 and +1. **d** Stereo view of recombinant HvExoI with G2SG-OMe (carbons: orange sticks) at 0.7 occupancy, bound across the +1 and putative +2 subsites. Water molecules are shown as red spheres. Separations of less than 3.50 Å from the active site residues (carbons: grey sticks) are shown as dashed lines. Derived |$2mF_{obs} − DF_{calc}$| electron density maps are contoured at 1σ for G2SG-OMe and Gol (blue mesh) in (**a–c**). In (**d**), electron density maps are contoured at 1σ for PEG (orange mesh) and Gol (green mesh) in the −1 subsite

anomeric proton of non-reducing Glc (H-1′) with intra-residual H-3′ and H-5′ (Supplementary Fig. 4; Supplementary Note 1) implied that the non-reducing β-D-glucopyranose ring of thio-cellobiose was bound to HvExoI in the $^4C_1$ conformation[32].

**Table 1 Binding ($K_D$) and inhibition ($K_i$) parameters for HvExoI by ligands and substrate analogues**

| Ligand/ inhibitor | $K_D{}^a/K_i{}^b$ (M) | $\Delta G^c$ (kJ × mol⁻¹) | Chemical formula[d] |
|---|---|---|---|
| Glc[a] | $0.16 \times 10^{-3}$ | −21.7 | $C_6H_{12}O_6$ |
| G6SG-OMe[a] | $0.008 \times 10^{-3}$ | −29.1 | $C_{13}H_{24}O_{10}S$ |
| 3dGlc[b] | $9.8 \times 10^{-3}$ | −11.7 | $C_6H_{12}O_5$ |
| 4dGlc[b] | $9.4 \times 10^{-3}$ | −11.8 | $C_6H_{12}O_5$ |
| Octyl-O-Glc[b] | $0.13 \times 10^{-3}$ | −22.6 | $C_{14}H_{28}O_6$ |
| Octyl-S-Glc[b] | $1.1 \times 10^{-3}$ | −17.2 | $C_{14}H_{28}O_5S$ |
| G2SG-OMe[b] | $2.55 \times 10^{-3}$ | −15.1 | $C_{13}H_{24}O_{10}S$ |

[a]Using Surface Plasmon Resonance with recombinant HvExoI
[b]Using inhibition kinetics with the 4-nitrophenyl β-D-glucopyranoside substrate[8] with native HvExoI
[c]Calculated according to $\Delta G = -RT \ln [K_i]$ or $\Delta G = -RT \ln [K_D]$[29]
[d]Chemical structures are shown in Supplementary Table 1

**Conformational Free Energy Landscape of Glc bound to HvExoI.** To further assess conformational states of bound Glc in the high-resolution structure of recombinant HvExoI in complex with Glc, quantum mechanics/molecular mechanics (QM/MM) metadynamics simulations were performed to asses Conformational Free Energy Landscape (FEL) maps of β-D-Glc bound in the active site of HvExoI (Fig. 5; Supplementary Fig. 2b). These analyses indicated that although Glc was engaged in extensive H-bond networks, the β-D-glucopyranose ring adopted $^4C_1$ chair and $^1S_3/B_{3,O}$ (−1 subsite) or $^4C_1$ chair and $B_{3,O}$ (+1 subsite) distorted skew boat/boat conformations (Fig. 5; left and middle panels). More precisely in the −1 subsite, H-bonds with Asp285 and Arg158 restrained the 2-OH group in the equatorial position, and stabilised the $^4C_1$ conformation by 8 kcal/mol more than that of $^1S_3$ or $B_{3,O}$, with a conformational energy barrier change of 11 kcal/mol. Conversely, hydroxyl groups of Glc in the +1 subsite interacted with Asp95, Arg158, Lys206, His207, Asp285, and Glu491, stabilising the $^4C_1$ Glc conformation by 2 kcal/mol more than that of $B_{3,O}$, with a conformational energy barrier change of 6 kcal/mol. Notably, these findings concurred with conformational FEL of isolated Glc[33], in which $^4C_1$ and $^1S_3/B_{3,O}$ conformers were most stable (Fig. 5; right panel). Nonetheless, FEL of Glc bound in the −1 and +1 subsites exhibited exclusively these two minima due to a restricted conformational space within the enzyme active site (Figs. 2a, 3b, 5) as demonstrated in other glycosyl hydrolases[34].

**Inhibition kinetics of substrate analogues in native HvExoI.** To shed light on the molecular basis of the Glc displacement route, we selected a series of deoxy-Glc (dGlc) and alkyl-glucoside derivatives, and substrate mimics that could potentially displace Glc[6]. To assess the strength of inhibition, we determined dissociation constants of enzyme-inhibitor complexes ($K_i$) for 3-deoxy-glucose (3dGlc), 4-deoxy-glucose (4dGlc), n-octyl β-D-glucopyranoside (octyl-O-Glc), n-octyl 1-thio-β-D-glucopyranoside (octyl-S-Glc), a hydrophilic polymer polyethylene glycol (PEG; putative fractional polysaccharide mimic[35]) and the thio-analogue methyl 2-thio-β-sophoroside (G2SG-OMe). All inhibitors showed competitive inhibition with $K_i$ values between $9.8 \times 10^{-3}$ M and $0.13 \times 10^{-3}$ M (Table 1). The most effective was octyl-O-Glc with $K_i$ of $0.13 \times 10^{-3}$ M, while surprisingly G2SG-OMe was weak with $K_i$ of $2.55 \times 10^{-3}$ M, while PEG was non-inhibitory, as expected.

**Glc is not removed from native HvExoI by dGlc derivatives.** To test the hypothetical Glc displacement route by using dGlc derivatives, crystals were perfused with saturating concentrations of

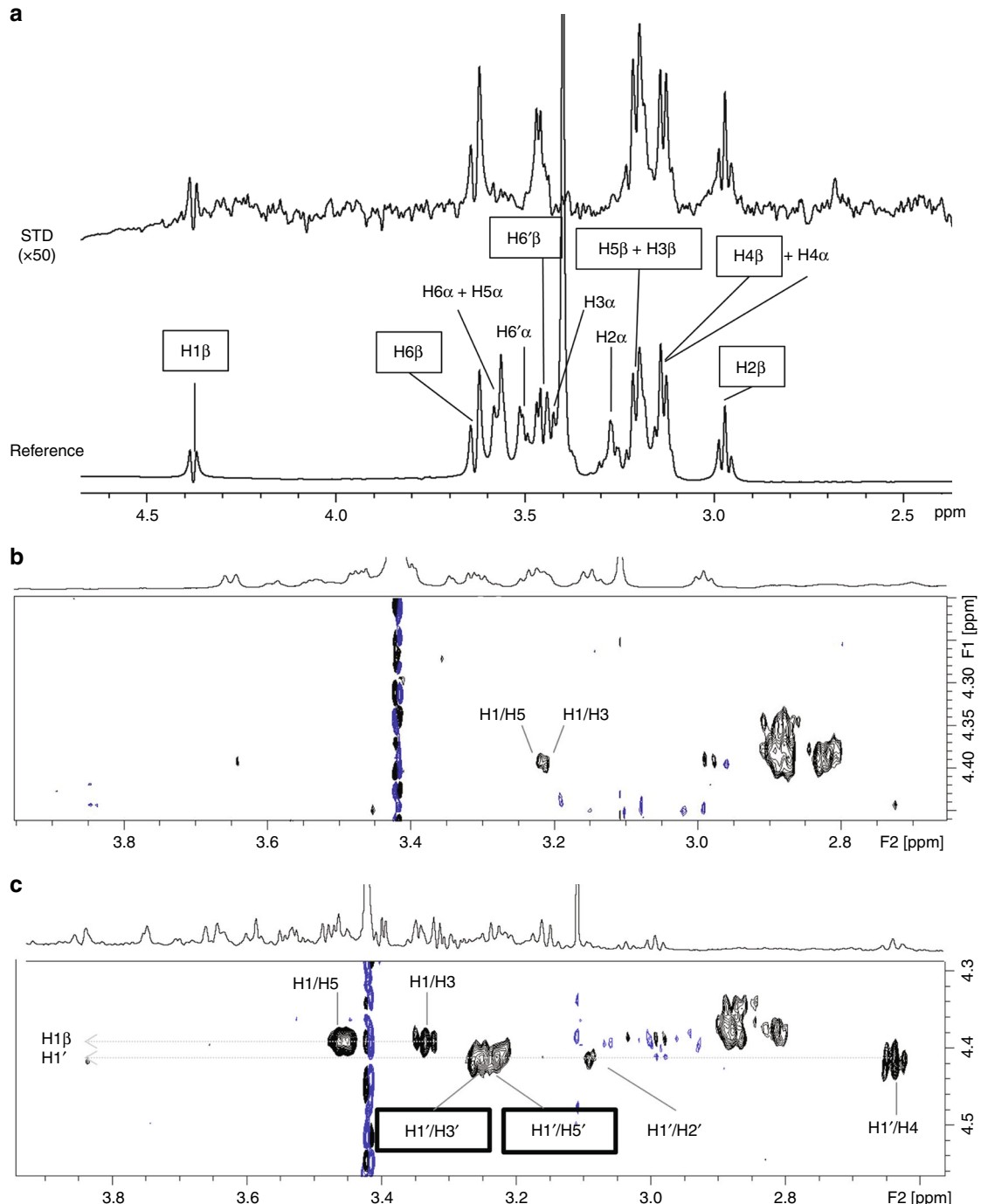

**Fig. 4** Recombinant HvExoI recognises β-D-Glc in the $^4C_1$ conformation. **a** $^1$H STD NMR spectrum (top) and reference spectrum (below), acquired with recombinant HvExoI and Glc (40 μM and 2.4 mM, respectively) at 600 MHz and 283 K. Only protons of β-D-Glc (squares) showed STDs. Off and on resonance frequencies were set at 100 ppm and 0.65 ppm, respectively. The Gaussian shaped pulse (30 ms) was used for selective irradiation with total saturation time of 2 s. **b** trNOESY spectrum of recombinant HvExoI and Glc (57 μM and 285 μM), acquired at 800 MHz, 283 K and with 300 ms mixing time. Only NOEs observed were weak trNOEs of Glc between H1/H3 and H1/H5. **c** trNOESY spectrum of recombinant HvExoI with Glc and thiocellobiose (57 μM, 285 μM, and 171 μM, respectively), acquired at 800 MHz, 283 K and with 300 ms mixing time. All NOEs were negative. trNOEs defining the $^4C_1$ conformation of non-reducing Glc of bound thiocellobiose are in boxes. Blue lines in (**b**) and (**c**) refer to residual noise signals

3dGlc (Fig. 2b; Supplementary Fig. 1b) and 4dGlc. The structures of both complexes showed that dGlc derivatives could not displace Glc from the −1 subsite, but that Glc consolidated to the −1 subsite. The well-defined electron density map indicated that 3dGlc was orientated parallel to Trp286 and Trp434 in the +1 subsite, through π-system stacking contacts, forming H-bonds

via C6-OH with Nε1 of Trp434 and Oε2 of Glu491, and via C4-OH with the N atom of Gly57 (Fig. 2b; Supplementary Fig. 1b). Notably, as 3dGlc rotated to a different position relative to that of Glc in native HvExoI (such that the C4-OH group of 3dGlc overlapped the position of C3-OH of Glc in the native structure), it could establish H-bonds with Gly57 and Glc in the −1 subsite.

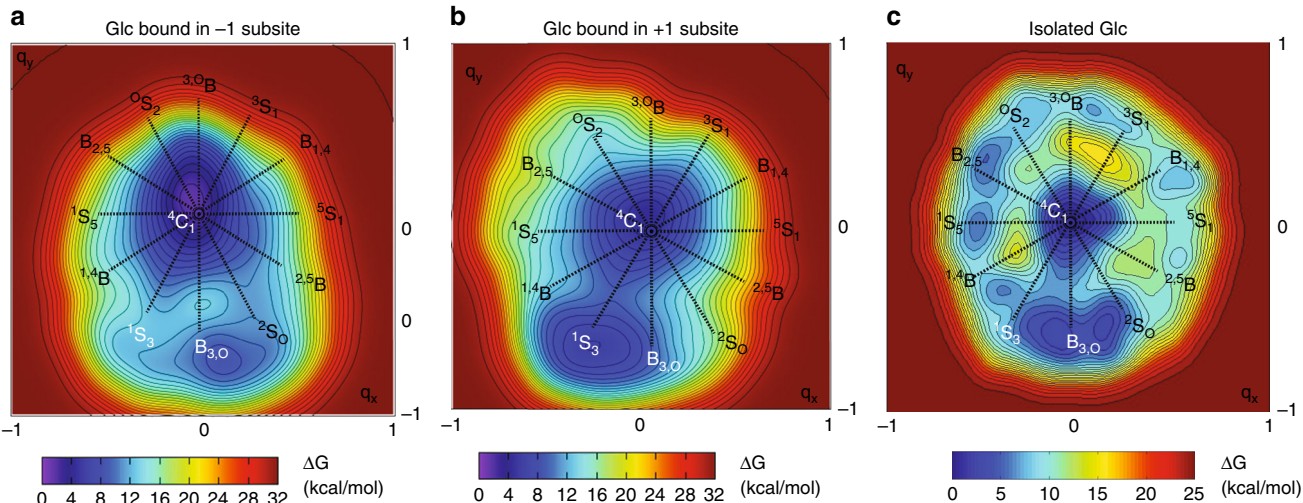

**Fig. 5** Conformational FEL maps of β-D-Glc bound in the active site of HvExoI. Conformational FEL of β-D-Glc in the −1 (**a**) and +1 (**b**) subsites are based on ab initio QM/MM metadynamics. **c** Conformational FEL map of isolated Glc[34] (reproduced with permissions of the Journal of American Chemical Society; all rights reserved; https://pubs.acs.org/doi/10.1021/jacs.5b01156). Energy level lines are separated by 1 kcal/mol

**Alkyl-glucosides and PEG remove Glc from the active site.** Contrary to what we observed with dGlc derivatives, n-octyl β-D-glucosides with tighter $K_i$ values (Table 1) replaced Glc and bound across the active site (Fig. 2c; Supplementary Fig. 1c). While aliphatic chains of alkyl β-D-glucosides were threaded through the Trp286 and Trp434 aromatic clamp and did not interact with a surrounding environment, Glc moieties established a network of 10–11 mono- and co-operative bi- and tridentate H-bonds, like those observed in native (with trapped Glc) or recombinant Glc-perfused HvExoI (Figs. 2, 3). Similarly, after applying PEG at saturating concentrations to native HvExoI crystals, the 1.80 Å PEG-perfused structure revealed that Glc was replaced by five water molecules occupying the positions of the C1-OH to C6-OH groups of Glc in the −1 subsite (Fig. 2d; Supplementary Fig. 1d). The PEG molecule bound between Trp286 and Trp434 in two alternate conformations made a water-mediated H-bond with Gly57, which was reminiscent of the bond formed between C4-OH of 3dGlc and Gly57. The second PEG molecule bound near Trp434 did not contact the enzyme, but instead interacted with the first PEG via water-mediated H-bonds (Fig. 2d; Supplementary Fig. 1d).

**G6SG-OMe and G2SG-OMe bind to HvExoI in two distinct modes.** To understand how (1,6)- and (1,2)-linked substrate thio-analogues displace the Glc product from the active site, we perfused recombinant crystals of HvExoI and observed that the sugars adopted different poses (Fig. 3c, d; Supplementary Fig. 2c, d). It is worth noting that G6SG-OMe binds to the enzyme with $K_D$ of $0.008 \times 10^{-3}$ M, whereas G2SG-OMe binds about 320-fold weaker ($K_i$ of $2.55 \times 10^{-3}$ M) (Table 1). We observed that in the 1.57 Å structure of HvExoI with G6SG-OMe, the ligand formed well-defined electron densities for both Glc moieties across the active site (Fig. 3c; Supplementary Fig. 2c), and that the 1.68 Å structure with G2SG-OMe (refined at 0.7 occupancy at both subsites) had a less-defined density for the ligand, which bound to the enzyme in a markedly different position (Fig. 3d; Supplementary Fig. 2d). For G6SG-OMe, we observed short H-bonds in the −1 subsite, which resembled those observed in the native (Fig. 2a; trapped Glc) or recombinant (Fig. 3a; perfused with Glc) enzymes. In contrast, G2SG-OMe could not slide past the +1 subsite in the pocket, likely due to a thio-glycosidic bond

rigidity, and hence it projected with the reducing-end Glc moiety into the solvent at the putative +2 subsite, while the non-reducing-end was held in the adjacent +1 subsite. For the −1 subsite, we assigned glycerol and PEG molecules that formed H-bonds with the G2SG-OMe non-reducing end that also established the H-bond with Gly57; the reducing moiety of G2SG-OMe interacted via C4-OH with Oε2 of Glu287 and via C3-OH with Nε1 of Trp286 (Fig. 3d; Supplementary Fig. 2d). In the G6SG-OMe structure we noted significant elevations of B-factor values in two loops (Thr214-Glu228 and Glu491-Asn498) near the catalytic site entry (Supplementary Note 2); these B-factor elevations were not seen in the G2SG-OMe structure.

**HvExoI-G2SG-OMe complex is an intermediate for Glc exit path.** We considered the HvExoI in complex with G2SG-OMe (PDB 6MD6) (Fig. 3d; Supplementary Fig. 2d) to embody the attributes of an intermediate enzyme-substrate complex, and the disposition of G2SG-OMe that of an incoming substrate. Hence, we used this complex for investigations of the hypothetical Glc displacement route via multi-scale molecular modelling employing docking[19] and MD simulations, followed by GaudiMM[20] and PELE[21] pathway calculations.

Respective reciprocal docking of the Glc product and the β-D-glucopyranosyl-(1,2)-D-glucose (G2OG) or β-D-glucopyranosyl-(1,3)-D-glucose (G3OG) substrates into HvExoI:G2OG-OMe or HvExoI:Glc complexes, combined with MD simulations indicated that the existence of ternary HvExoI:Glc:G2OG or HvExoI:Glc:G3OG complexes (designated 1–6) was plausible, where Glc was bound in the −1 subsite, and G2OG or G3OG were attached at +1 and putative +2 subsites (Fig. 6a; Supplementary Figs. 5, 6, 8, Supplementary Notes 3–4, Supplementary Data 1). This suggested that if the incoming substrate binds at the +1 to +2 subsites, while Glc is still trapped in the −1 subsite, an alternative Glc exit path, other than through the +1 subsite needs to be considered. We also searched for substrate binding modes different to those at the +1 to +2 subsites, using docking of the G2OG or G3OG substrates in structures based on MD simulations of the native HvExoI:Glc complex (Supplementary Figs. 5–8; Supplementary Note 4; Supplementary Data 1–2). However, no stable binding sites were found other than those at +1 and +2 subsites. In addition, docking and MD simulations

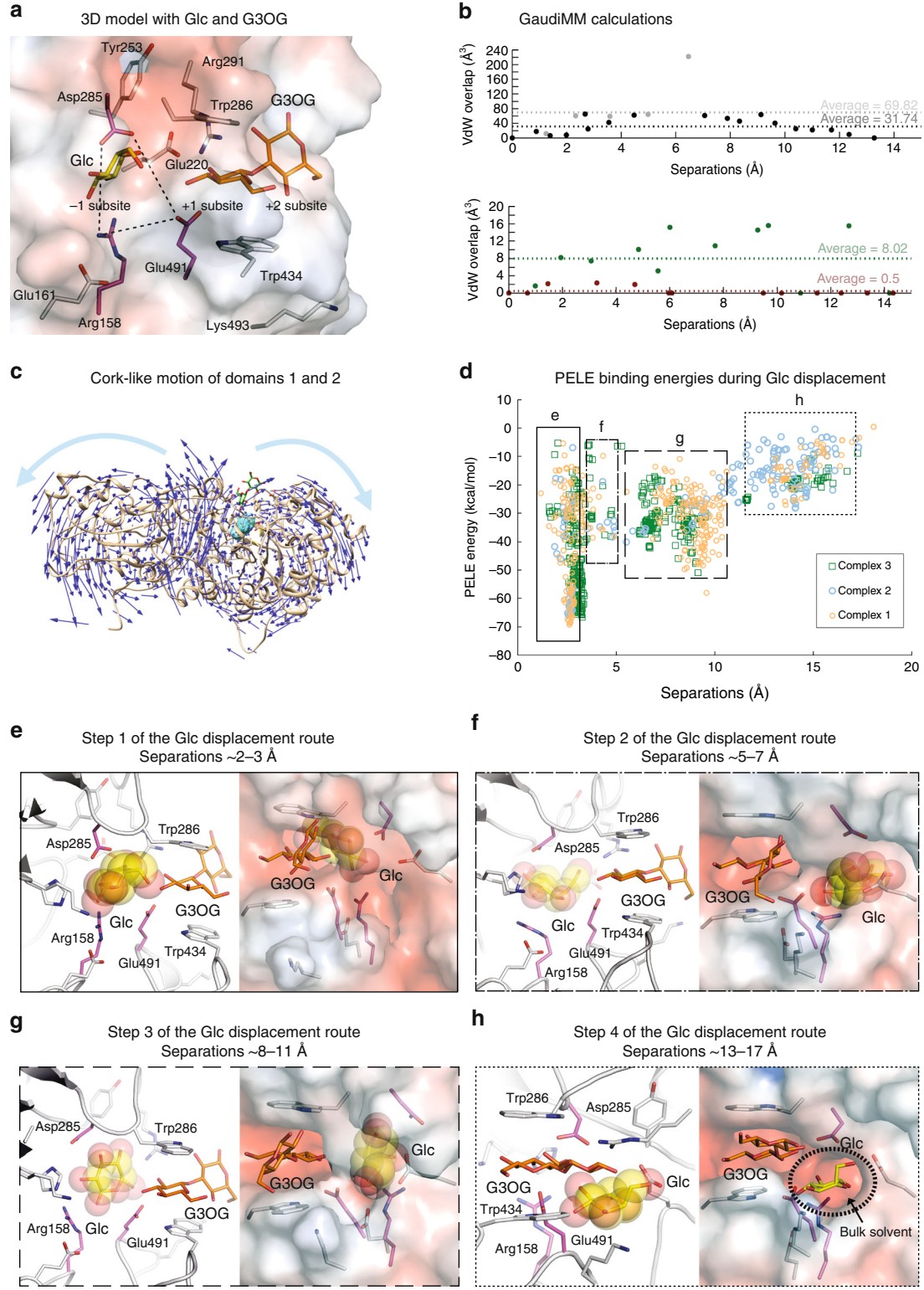

**a** 3D model with Glc and G3OG

**b** GaudiMM calculations

**c** Cork-like motion of domains 1 and 2

**d** PELE binding energies during Glc displacement

**e** Step 1 of the Glc displacement route
Separations ~2–3 Å

**f** Step 2 of the Glc displacement route
Separations ~5–7 Å

**g** Step 3 of the Glc displacement route
Separations ~8–11 Å

**h** Step 4 of the Glc displacement route
Separations ~13–17 Å

unequivocally revealed that the HvExoI:Glc:G2OG and HvExoI:Glc:G3OG complexes (Fig. 6a) converged and were stable, and could be used for investigations of Glc displacement. MD simulations also revealed that stable binding of substrates required Trp434 adopting the parallel orientation to that of Trp286, where C-H/π interactions mediated binding. Notably, binding of the G2OG or G3OG substrates triggered the conformational change of Tyr253, modifying the buried lateral

cavity adjacent to the −1 and +1 subsites. We observed that when only one Glc was bound at the −1 subsite, Tyr253 remained H-bonded to the carbonyl oxygen of the Trp286 backbone for the full extent of simulation. The sidechains of the catalytic nucleophile Asp285 and acid/base Glu491, together with Arg158 (Fig. 6a; dashed lines) physically separated this cavity from the −1 subsite. In complexes 1, 3 and 4, the Tyr253 side chain moved away from this lateral cavity enlarging it, while in complex 2,

**Fig. 6** Displacement pathway of Glc, based on crystal structures and molecular modelling. **a** Converged HvExoI:Glc:G3OG complex obtained by docking. Selected residues (carbons: atomic and purple sticks), Glc (carbons: yellow sticks) and G3OG (carbons: orange sticks) are shown. Glc in the −1 subsite is separated from the lateral cavity by the Arg158-Glu491-Asp285 (carbons: purple sticks) toll-like barrier (dashed lines). Surface morphology coloured by electrostatic potentials: white, neutral; blue, +5 kT e$^{-1}$; red, −5 kT e$^{-1}$. **b** Glc cannot exit, when crystallographic coordinates are considered by GaudiMM (grey symbols). Glc displacement only occurs when protein atoms are relocated to the lowest-energy normal mode (black). Clashes of Glc are shown as the function of separations from the Glc geometric centre in the −1 subsite. Displacement routes present fewer clashes, when MD structures are considered (green) and protein atoms move (dark red). **c** Lowest-energy normal mode shows a cork-like motion of domains 1 and 2. **d** Glc displacement routes calculated by PELE using ternary HvExoI:Glc:substrate complexes 1–3. PELE binding energies, plotted as a function of Glc separation from the −1 subsite, show that Glc traverses via the lateral cavity at the separation of 8 Å from protein surface. Full, semi-dashed, dashed and dotted lines specify separation ranges of 2–3 Å, 5–7 Å, 8–11 Å and 13–17 Å (referring to line thickness in (**e**–**h**)) between respective centres of masses and C4-OH groups of Glc in initial and final structures. **e**–**h** Four steps along the Glc displacement route based on complexes 1–3 (Glc carbons: yellow spheres). Left and right images are rotated by 200° (**e**–**g**) and −50° (**h**) along y-axes. **e** Glc is in the −1 subsite. **f** Movements of Arg158, Asp285 and Glu491 allow Glc traversing into the lateral cavity. **g** Glc in the lateral cavity is partly exposed to the bulk solvent. **h** Glc is fully exposed to the bulk solvent. Black dotted ellipsoid indicates the transient solvent-exposed aperture of the lateral cavity that facilitates Glc displacement

this side chain rotated to the bottom of the cavity making it shallower (Supplementary Figs. 6, 8; Supplementary Notes 3–4).

To find out if binding of substrates by HvExoI lacking the bound Glc product would lead to tighter or weaker binding than that with Glc included, we carried out further calculations and compared binding affinities. Docking calculations of G2OG, G3OG and β-D-glucopyranosyl-(1,6)-D-glucose (G6OG) disaccharide substrates at the −1 and +1 subsites, predicted binding with higher affinities (Goldscore scoring function values of 66 for G2OG, 76 for G3OG and 74 for G6OG), when the Glc product was absent in the −1 subsite. However, when the Glc product was included in the −1 subsite, docking of G2OG, G3OG and G6OG in the +1 and putative +2 subsites predicted binding with lower affinities (Goldscore scoring function values of 60 for G2OG, 57 for G3OG and 61 for G6OG). These data indicated that bound Glc lowered binding energies for incoming substrates as they had no access to the higher affinity −1 subsite.

**Discovery of substrate-product assisted processivity in HvExoI.** Initially, we used the GPathFinder extension of the GaudiMM platform[20] to reveal potential Glc exit routes from the active site into the bulk solvent, whereby we considered steric clashes. The initial set of simulations was performed with the HvExoI:Glc:G2OG ternary complex 1, based on the coordinates of HvExoI in complex with G2SG-OMe and Glc docked in the −1 subsite. No exit path was detected in this case, as Glc never overcame steric clashes to vacate the −1 subsite (Fig. 6b; top panel; Supplementary Data 3). However, further calculations with the protein backbone displaced by 2.2 Å along the lowest-energy normal mode strikingly identified the Glc displacement path. As a bonus, these calculations also revealed a cork-like motion between domains 1 and 2, which was correlated with the open and closed states of the active site (Fig. 6c). The same ligand path was also obtained with the MD-derived HvExoI:Glc:G3OG ternary complex 3 (Fig. 6b; bottom panel), highlighting the importance of incorporating local and collective protein motions in these calculations.

To explore the Glc displacement route in detail, the ligand migration calculations of the PELE approach[21] were used, whereby structures generated at each step along the path (by sequential ligand and protein geometric perturbations followed by energy minimisation using the OPLS-AA force field) were evaluated and accepted or rejected according to a Metropolis criterion at a given temperature. Using this approach, the HvExoI:Glc:G2OG and HvExoI:Glc:G3OG complexes 1–3 converged, and revealed that Glc egressed from the −1 subsite to the adjacent lateral cavity that was enlarged after Tyr253 rotated. This cavity was defined by an ensemble of 14 residues: Trp156, Arg158, His207, Phe208-backbone, Asp211, Asn219, Glu220,

Ser252, Tyr253-backbone, Ser254, Asp285, Arg291, Glu491 and Thr492 (Fig. 6; Supplementary Fig. 9). This cavity was partially separated from the solvent by the Asn219, Glu220 and Arg291 sidechains, with the two latter residues forming a salt bridge with Oε1 and Oε2 of Glu220 to NH1 of Arg291 at separations of 3.10 Å/2.65 Å, 2.77 Å/3.56 Å and 4.65 Å/2.84 Å for complexes 1, 2, and 3, respectively. The abundance of data points in the PELE plot at separations between 8–11 Å (Fig. 6d) indicated that there was a local energy minimum for Glc binding in this cavity (Fig. 6d; subpanel g). For complex 2, PELE calculations suggested a more transient passage of Glc through the lateral cavity, most likely due to a shallower profile resulting from the specific Tyr253 conformation, compared to other complexes. Nevertheless, in all cases, if Glc was to advance to the lateral cavity from the −1 subsite, it must traverse the space between Asp285, Glu491 and Arg158, which likely represent a toll-like barrier (Fig. 6a, dashed lines; Fig. 6e); this Glc passage corresponded to fewer data points at separations between 5–7 Å (Fig. 6d; Supplementary Data 4). With Glc bound in the −1 subsite, the shortest separations between the Arg158-Asp285, Arg158-Glu491 and Glu491-Asp285 sidechains were those at 5.10 Å/5.42 Å/5.17, 3.09 Å/3.16 Å/4.42 Å and 5.39 Å/6.32 Å/6.65 Å in respective complexes 1, 2 and 3. Notably, conformational changes of Arg158, Asp285 and Glu491 sidechains facilitated Glc movement (Fig. 6e, f), and altered the separations specified above to 7.07 Å/5.33 Å/6.21 Å, 3.00 Å/2.97 Å/6.82 Å and 6.96 Å/7.23 Å/7.71 Å, when Glc bypassed the toll-like barrier. Once in the lateral cavity, Glc was free to exit from the cavity into the bulk solvent via a transient and autonomous aperture. This transient opening was formed through rotations and backbone fluctuations of Glu220, Arg291, Thr492 and Lys493, and surrounding residues in the vicinity of the bound β-D-glucoside molecule (Fig. 6h). As Glc migrated across the lateral cavity (Fig. 6e, f), it established H-bonds[36] with the protein residues and incoming substrates to maintain energetic favorability (Supplementary Tables 2–4). This suggested that the hydrophilic environment of the toll-like barrier and the lateral cavity may have evolved for this exit route and may be evolutionarily conserved in GH3 enzymes (Supplementary Fig. 9). Finally, and most importantly, the Glc displacement route raised the possibility that this trajectory may facilitate processive catalysis in HvExoI and other GH3 exo-hydrolases with pocket-shaped active sites.

**Proof of concept using the R158A/E161A variant.** Ligand migration calculations using PELE (Fig. 6) and the conservation patterns of the toll-like barrier (Supplementary Fig. 9) led us designing the double non-conservative R158A/E161A variant to critically asses the roles of these residues in binding Glc and the

**a** Variant R158A/E161A in ligand-free form

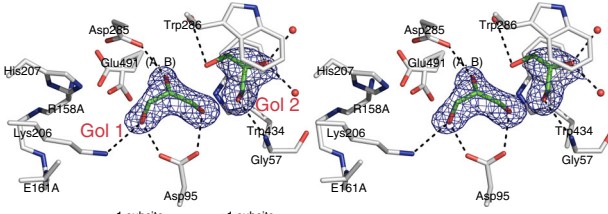

**b** Variant R158A/E161A perfused with Glc

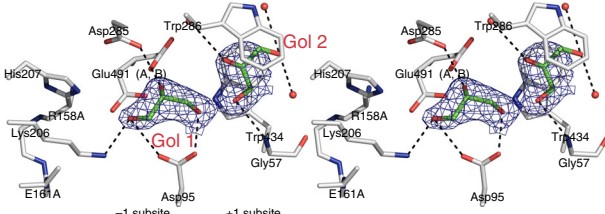

**c** Variant R158A/E161A perfused with G6SG-OMe

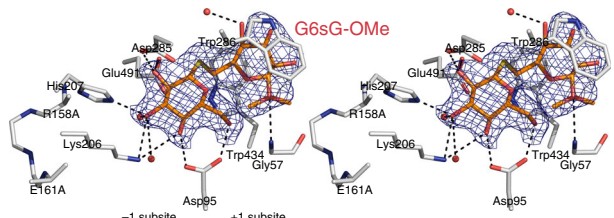

**Fig. 7** R158A/E161A of HvExoI and with perfused Glc or G6SG-OMe. **a** Stereo view of a ligand-free form of R158A/E161A. Two glycerol molecules (carbons: green sticks) at 1.0 occupancies are bound in the −1 and +1 subsites. **b** Stereo view of R158A/E161A with perfused Glc, which failed to bind, instead two glycerol molecules (carbons: green sticks) are bound at 1.0 occupancies in the −1 and +1 subsites. **c** Stereo view of R158A/E161A with G6SG-OMe (carbons: orange sticks) at 1.0 occupancy, bound across the −1 and +1 subsites. Separations of less than 3.30 Å from electronegative atoms of active site residues (carbons: grey) and water oxygens, are shown as dashed lines. Derived $|2mF_{obs} - DF_{calc}|$ electron density maps are contoured at 1σ (blue mesh)

G6SG-OMe substrate thio-analogue (Fig. 7; Supplementary Fig. 10). The 1.65 Å and 2.21 Å structures without and with perfused Glc showed two glycerol molecules, one each in the −1 and +1 subsites (Fig. 7a, b; Supplementary Fig. 10a, b), and the absence of Glc despite perfusing crystals at saturating concentrations. On the other hand, the 2.30 Å structure of R158A/E161A perfused with G6SG-OMe strikingly revealed this sugar bound across the active site, but in a different pose compared to that of the wild-type (WT) enzyme (cf. Figures 3c, 7c; Supplementary Figs. 2c and 10c). In the variant structure the position of the Glc moiety in the −1 subsite matched that of WT, but the disposition of the reducing-end Glc moiety was flipped in its position between the indole moieties of Trp286 and Trp434, such that intra-ring oxygen of the β-D-glucopyranose moiety pointed to Trp434. Obviously, this G6SG-OMe pose was due to Ala replacements for Arg158 and Glu161.

## Discussion

Substrates and products in enzyme active sites bind and unbind at fast rates that challenge investigations of their trajectories, leading to the lack of a deep understanding of these hallmarks of enzyme catalysis[37–39]. Here, we capitalise on the Glc product entrapment

observation in native HvExoI isolated from seedlings[6], which renders it an archetype model to examine product/substrate dissociations/associations in a pocket-shaped active site. Most approaches towards the descriptions of reactant movements are based on MD simulations of substrate and product binding and unbinding, including random collisions or diffusion[40]. Other studies reveal substrate/product migration pathways that accompany conformational changes of protein backbones or secondary structures[41,42].

When we first described the HvExoI structure[6–9], we were unable to explain why the trapped Glc product remained bound in the active site and has not diffused away, and what was the implication of this unassuming observation. The obvious explanation for not seeing naturally bound products or co-factors in structures of other GH enzymes is that these proteins are generated in recombinant hosts, where intracellular concentrations of potential enzyme reactants are not high enough during protein maturation. To this end, solving crystal structures of enzymes purified from native sources offers an additional information that could be beneficial for the understanding of catalytic cycles.

When we first detected the Glc product entrapment in native HvExoI, we hypothesised that the role of Glc may be linked to the pre-organised state of the active site to maintain efficient catalysis required for the growth of a plant embryo[9]. One way of preserving this pre-organised state would be to keep entropic costs of catalysed reactions low, where an entrapped product from a previous hydrolytic cycle could lower overall entropic demands for binding of incoming substrates[1,43]. The Glc product retention in the active site may also disfavour binding of incorrect substrates through product dissociation rates that would govern the selectivity of substrate binding[44]. Others may argue that although most enzymes have evolved to use conformational adjustments to favour tight binding of correct substrates and product release, some enzymes may have not acquired this asset. This assumption was confirmed by perfusing dGlc derivatives that akin of reaction products could not remove Glc, but substrate analogues and mimics could.

As mentioned above, despite the plethora of structural data for HvExoI[6–9,15–17], we could not explain, how Glc egresses from the 13 Å-deep pocket-shaped active site, although we detected crosstalk between residues entrapping Glc in the −1 subsite, and those in the +1 subsite delineated by Trp286 and Trp434. Function of Trp residues in retaining Glc or substrates is unsurprising, as these residues have extended heterocyclic indole ring systems with amphipathic characteristics, high de-localised electron densities and permanent dipole moments due the intra-ring nitrogens, that form H-bonds[29]. In all 22 structures of HvExoI we so far resolved (Supplementary Table 5), Trp286 and Trp434 at the +1 subsite showed no structural heterogeneity, and at separations of about 4 Å from ligands were compliant with the provision of two sets of C-H/π interactions[45] depending on the stereochemistry of bound ligands[7,17]. Crystalline HvExoI was also perfused with the (3)-β-D-Glc-S-(1,3)-β-D-Glc-(1,)$_{14–22}$ polymeric substrate[46], but we did not observe potential binding sites beyond the −1 and +1 subsites.

The time-space averaged vision of crystal structures presented here was supported by the multi-scale molecular modelling, based on docking[19], MD simulations, QM/MM, GaudiMM[20] and PELE[21], that provided the first clear view of succession of events during Glc displacement. This approach also hinted that trapped Glc may play a surprising role in catalysis and led to the discovery of processivity by this exo-hydrolase. In the light of this discovery we coin the term 'substrate-product assisted processive catalysis', due to the key role that a substrate and a product play in the evocation of the catalytic pathway.

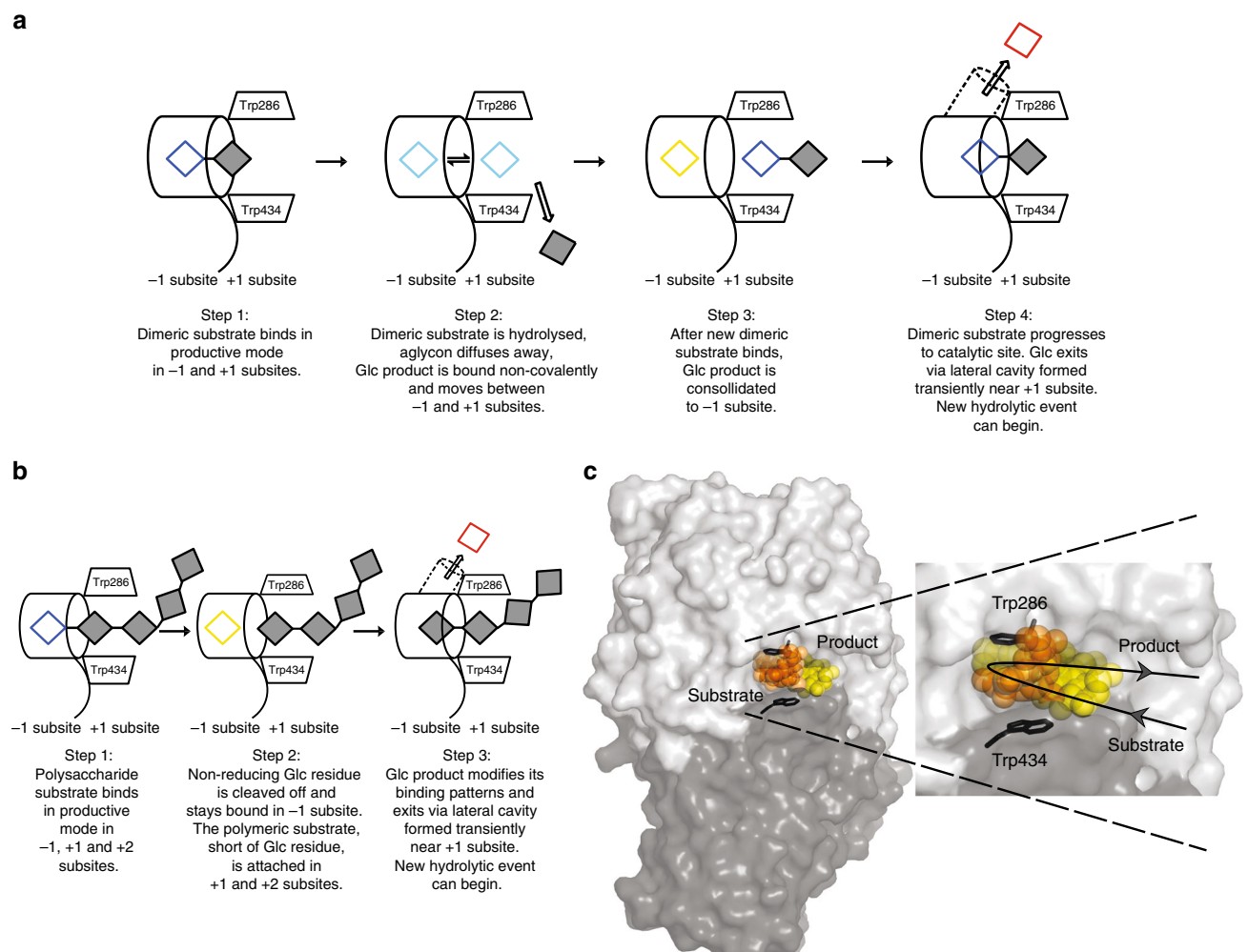

**Fig. 8** The mechanism of substrate-product assisted processive catalysis by HvExoI. **a** Mechanism of Glc displacement with a disaccharide. After the disaccharide (empty blue and filled grey squares) bound in −1 and +1 subsites (step 1) is hydrolysed and an aglycon diffuses away (step 2), Glc (cyan square) remains non-covalently trapped and oscillates (double arrow) between the −1 and +1 subsites. Glc (yellow square) is consolidated to the −1 subsite after an incoming substrate binds (step 3), which later advances to the catalytic site. Glc (red square) modifies its binding patterns and exits (large arrow) via an autonomous and transient lateral cavity (cylinder in dashed lines) formed near the catalytic site (step 4); a next hydrolytic cycle can begin. **b** Mechanism of Glc displacement with a polysaccharide to facilitate substrate-product assisted processive catalysis. After non-reducing (blue square) and penultimate (filled square) residues bind in a productive mode at the −1 and +1 subsites (step 1), the non-reducing Glc residue (yellow square) is cleaved off, with the remainder of the substrate attached (step 2). Glc (red square) modifies its binding patterns and is released (large arrow) via a lateral cavity (cylinder in dashed lines, step 3). Here, the next hydrolytic cycle continues with the same polysaccharide, where the polysaccharide, short of one Glc (filled squares) advances into the catalytic site after uninterrupted binding. Non-reducing (blue) and hydrolysed (cyan-yellow-red in (**a**), yellow-red in (**b**)) Glc is shown. **c** Structural basis of substrate-product assisted processive catalysis. Left: The substrate (orange spheres) slides into the active site (protein surface: domain 1-light grey, domain 2-dark grey). After hydrolysis, the Glc product (yellow spheres) exits from the −1 subsite through the lateral cavity into the bulk solvent. Right: detail of the active site. Circle line with arrows indicates directionality during processive catalysis. The image was generated using the coordinates of HvExoI:G6SG-OMe and HvExoI:G2SG-OMe complexes (Fig. 3c–d; orange spheres), and the Glc displacement snapshots based on PELE calculations (Fig. 6e–h; yellow spheres)

We demonstrate that the substrate binding and product displacement routes proceed through stages that are carefully orchestrated in succession (Fig. 8). We suggest that a disaccharide (Fig. 8a) binds near the catalytic site at the +1 and putative +2 subsites most likely through random collisions, from where it progresses to the −1 and +1 subsites. Following binding in a productive mode (step 1), the disaccharide (empty blue and filled grey squares) is poised for hydrolysis. After the non-reducing Glc is cleaved off and aglycon diffuses away, the Glc product is entrapped non-covalently, whereby it may oscillate between the −1 and +1 subsites (cyan square) (step 2). After the next disaccharide binds at +1 and +2 subsites, the Glc product

(yellow square) is consolidated to the −1 subsite. This is linked to Tyr253 rotation, which enlarges the lateral cavity adjacent to the −1 and +1 subsites (step 3). Next, the Glc product (red square) traverses from the −1 subsite through rotations of the Arg158 and Asp285 sidechains and associated backbone atoms, and bypasses the toll-like barrier shaped by Asp285, Glu491 and Arg158, via this autonomous lateral cavity (step 4, cavity shown as a barrel in dashed lines). This cavity is formed transiently near the +1 subsite, through which the Glc product exits (arrow) via an aperture into the bulk solvent (Fig. 8a; step 4). Strikingly, when the toll-like barrier is disrupted in the R158A/E161A variant, Glc no longer binds in the active site (Fig. 7), which underlines the

vital role of the barrier in Glc entrapment. The sequence of events involving incoming substrate binding and Glc displacement are captured in Supplementary Movie 1.

We propose a similar substrate/product progression route (Fig. 8b) takes place with a polysaccharide (empty blue and filled grey squares), often with differently linked β-D-glucosyl residues[8,19], through processive catalysis. Here, (step 1) after Glc (blue square) is cleaved off from the polysaccharide, (step 2) the Glc product (yellow square) remains bound at the −1 subsite, while the remainder of the polysaccharide (grey squares) stay attached at the +1 and putative +2 subsites. As the Glc product (red square) exits (arrow) through the lateral cavity (step 3, shown as a barrel in dashed lines), the polysaccharide can advance to the −1 and +1 subsites, such that an uninterrupted hydrolysis of the polymer continues until it is completely hydrolysed. In both instances, after Glc egresses (Fig. 8a, b), the next hydrolytic cycle that is facilitated by the dedicated substrate binding and product displacement routes, can continue. Consequently, this coordinated system with oligo- and polymeric substrates, resembles a loop or a conveyer belt that directionally shuffles substrates and ensures efficient catalysis (Fig. 8c). Although in all computations, we modelled Glc in $^4C_1$ conformation, we do not rule out that the Glc distortion to a higher energy of the $^1S_3$ conformer, as it occurs in crystal structures, may contribute to displacement of the Glc product or its progression.

To our knowledge, this type of processive catalysis[47–49] has not been described in any exo-acting hydrolase with a pocket-shaped active site like that of HvExoI. In this work, we define processive catalysis in broader terms, rather than those based on hydrolytic profiles of products[50]. Considering that HvExoI operates on polysaccharides such as (1,3;1,4)-β-D-glucans contained in plant cell walls[9,51], this finding is unsurprising, as the enzyme's efficient hydrolysis is the prerequisite for seed germination and the development of a plant embryo[9,23,51]. Processive catalysis (non-dissociative sequential degradation) has been accepted for endo-acting hydrolases with cleft- or groove-shaped catalytic sites, such as (1,3)- and (1,3–1,4)-β-D-glucan endohydrolases[52], and (1,4)-β-D-glucan endohydrolases[53], and for exo-acting chitinases[50] and cellobiohydrolases[54] with ridge- or tunnel-shaped sites. Conversely, β-D-glucosidases and exo-acting hydrolases with funnel-, crater- or pocket-shaped[6,55] active sites are deemed to be non-processive[48,49].

Our concept of substrate and product processive co-operative routes with pocket-shaped exo-hydrolases[6–9] is associated with chemical signalling, where entrapped products, bound substrates and enzymes co-operatively create pathways dedicated to product dissociation. The hallmark and advantage of processive catalysis is a high efficiency, where oligo- or polysaccharides are directionally threaded without the catalyst losing contact with substrates to keep binding entropic cost low, while performing multiple hydrolytic events on the same substrate molecule. While this work was focussed on a plant GH3 hydrolase, we suggest that substrate-product assisted processive catalysis may be more prevalent among exo-hydrolases with pocket-shaped active sites, irrespective of their substrate specificity. To this end, our findings could promote further investigations of enzymes involved in biomass degradation, where one of the challenges is to circumvent the bottleneck of product inhibition, a problem that could be addressed through the better understanding of enzyme catalytic cycles. More broadly, substrate-product assisted processive catalysis in enzymes with pocket-shaped catalytic sites is likely to have significance in other enzymes due to the plasticity of protein structures. It is exactly this plasticity that accounts in HvExoI for product dissociation, although in other enzymes the atomic details of reactants movements may be different.

The discovery of substrate-product assisted processive catalysis in HvExoI prompts investigations of the evolutionary origin of this mechanism. The availability of the presented experimental and computational tools alongside a rich source of information for the GH3 family enzymes that originate from all phylla[9,18] (currently around 23,000 entries), will now allow the study of evolution of this catalytic mechanism. Preliminary analyses of 500 sequences related to HvExoI revealed that the residues underlying this mechanism are conserved in land plants, but absent in red and green algae, suggesting that this mechanism is rather ancient and has evolved in land plants about 470 million years ago.

We conclude that through the descriptions of associative and dissociative reactant trajectories that we explored using our interdisciplinary approach, we are now in a better position to understand how reactants advance in active sites, and how to improve enzyme catalytic rates, stability, product inhibition and drug discovery.

## Methods

**Materials.** 3-Deoxy-glucose (3dGlc) and 4-deoxy-glucose (4dGlc) were obtained from Glycoteam GmbH (Hamburg, Germany). N-octyl β-D-glucopyranoside (octyl-O-Glc) and n-octyl 1-thio-β-D-glucopyranoside (octyl-S-Glc) were from Anatrace Inc. (Maume, OH, USA). 4-Nitrophenyl β-D-glucopyranoside (4NPGlc), glycerol, myo-inositol and other chemicals were obtained from Sigma-Aldrich (USA) or as described previously[6,7,17]. $CH_3CN$ was from BDH Laboratory Supplies (UK), and polyethylene glycol 400 (PEG400; $n = 5$–10) was from Fluka Biochemica (Germany).

**Synthesis of methyl 6-thio-β-gentiobioside.** The synthesis proceeded in two steps as follows.

First, methyl 6-thio-β-gentiobioside heptaacetate (chemical structure in Supplementary Table 1): 2,3,4,6-Tetra-O-acetyl-1-thio-β-D-glucopyranose (845 mg, 2.32 mmol)[56], 1,4-dithioerythritol (357 mg, 2.32 mmol) and cysteamine (1179 mg, 2.32 mmol) were successively added to a solution of methyl 2,3,4-tri-O-acetyl-6-deoxy-6-iodo-β-D-glucopyranoside (1 g, 2.32 mmol) in hexamethylphosphoramide (HMPA) (7 mL)[57]. The mixture was kept for 1 h at room temperature then precipitated into ice-water (100 mL). The solid was collected on Celite, washed with water and then dissolved in $CH_2Cl_2$. The organic phase was washed with water, dried over $Na_2SO_4$ and concentrated. Column chromatography in silica gel (EtOAc-light petroleum 1:1) afforded methyl 6-thio-β-gentiobioside heptaacetate (1.3 g, 84% yield). $[\alpha]_D^{25}$ -20.0 (c 0.73, $CHCl_3$); ESI-HRMS(+) calcd for $C_{27}H_{38}O_{17}NaS$ 689.17219; found 689.17081; $^1H$ NMR ($CDCl_3$): δ 5.16 (t, 1 H, $J$ = 9.5 Hz), 5.15 (t, 1 H, $J$ = 9.5 Hz), 5.0 (t, 1 H, $J$ = 9.8 Hz), 4.95 (t, 1 H, $J$ = 10.0 Hz), 4.93 (dd, 1 H, $J$ = 8.0 Hz, $J$ = 10 Hz), 4.88 (t, 1 H, $J$ = 9.5 Hz), 4.65 (d, 1 H, $J$ = 10.0 Hz), 4.39 (d, 1 H, $J$ = 8.0 Hz), 4.22 (dd, 1 H, $J$ = 5.0 Hz, $J$ = 12.5 Hz), 4.12 (dd, 1 H, $J$ = 2.0 Hz, $J$ = 12.5 Hz), 3.65 (m, 2 H), 3.51 (s, 3 H), 2.79 (m, 2 H), 2.08–1.97 (m, 21 H); $^{13}C$ NMR ($CDCl_3$): δ 169.6–169.3, 101.6, 83.9, 76.0, 74.4, 73.8, 72.7, 71.9, 71.3, 70.2, 68.2, 62.1, 57.1, 31.2, 20.77–20.72.

Second, methyl 6-S-β-D-glucopyranosyl-6-thio-β-D-glucopyranoside (methyl 6-thio-β-gentiobioside; G6SG-OMe) (chemical structure in Supplementary Table 1): sodium methylate 1 M (300 μL) was added to a solution of methyl 6-thio-β-gentiobioside heptaacetate (300 mg, 0.45 mmol) in MeOH (30 mL). After 4 h at room temperature, the mixture was neutralised with Amberlite IR 120 $H^+$ resin, filtered and concentrated. The residue was dissolved with water, and freeze dried affording methyl 6-thio-β-gentiobioside (166 mg, 99% yield). $[\alpha]_D^{25}$ -26.2 (c 1.13, $H_2O$); ESI-HRMS(+) calcd for $C_{13}H_{24}O_{10}NaS$ 395.09824; found 395.09750; $^1H$ NMR ($D_2O$): δ 4.69 (d, 1 H, $J$ = 10.0 Hz), 4.43 (d, 1 H, $J$ = 8.0 Hz), 3.94 (dd, 1 H, $J$ = 2 Hz, $J$ = 12.5 Hz), 3.75 (dd, 1 H, $J$ = 5.5 Hz, $J$ = 12.0 Hz), 3.68 (m, 1 H), 3.62 (s, 3 H), 3.54–3.24 (m, 8 H), 2.97 (dd, 1 H, $J$ = 8.0 Hz, $J$ = 14.5 Hz); $^{13}C$ NMR ($D_2O$): δ 103.3, 86.0, 79.8, 77.2, 76.0, 75.5, 73.2, 72.5, 72.4, 69.5, 60.9, 57.4, 31.3.

**Synthesis of methyl 2-thio-β-sophoroside.** The synthesis proceeded in three steps as follows.

First, methyl 3,4,6-tri-O-acetyl-2-O-trifluoromethanesulfonyl-β-D-mannopyranoside (chemical structure in Supplementary Table 1): Distilled pyridine (0.5 mL) and trifluoromethanesulfonic anhydride (113 μL, 3.6 equiv) were successively added to an ice-cold solution of methyl 3,4,6-tri-O-acetyl-β-D-mannopyranoside[58] (60 mg, 0.19 mmol) in anhydrous $CH_2Cl_2$ (3 mL). The mixture was stirred at 0 °C for 30 min and at room temperature for 1 h. The solution was diluted with water and extracted with $CH_2Cl_2$. The organic phase was washed with aqueous $KHSO_4$ (20%, v/v), saturated aqueous $NaHCO_3$, dried over anhydrous $Na_2SO_4$ and concentrated. The crude triflate (84 mg) was obtained in a quantitative yield and directly used without purification.

Second, methyl 3,4,6-tri-O-acetyl-2-S-(2,3,4,6-tetra-O-acetyl-β-D-glucopyranosyl)-2-thio-β-D-glucopyranoside (methyl 2-thio-β-sophoroside heptaacetate) (chemical structure in Supplementary Table 1): diethylamine (1 mL) under Argon was added to a stirred solution of 2,3,4,6-tetra-O-acetyl-1-thio-β-D-glucopyranose[59] (91 mg, 0.22 mmol), to which crude methyl 3,4,6-tri-O-acetyl-2-O-trifluoromethanesulfonyl-β-D-mannopyranoside (84 mg, 0.19 mmol) in anhydrous DMF (10 mL) was added. The mixture was stirred overnight at room temperature and then concentrated. A solution of the residue in $CH_2Cl_2$ was washed in water, dried over anhydrous $Na_2SO_4$ and concentrated. Flash chromatography on silica gel (ethyl acetate/petroleum ether, 1:1) afforded methyl 2-thio-β-sophoroside heptaacetate (26 mg, 21%). Analytical data agreed with those reported previously[60]. ESI-MS: m/z = 689 [M$^+$Na]$^+$; $^1$H NMR (CDCl$_3$) δ 5.12 (t, 1 H, J = 9.3 Hz), 5.04 (t, 1 H, J = 9.4 Hz), 4.97 (m, 3 H), 4.73 (d, 1 H, J = 10.2 Hz), 4.36 (d, 1 H, J = 8.5 Hz), 4.30 (dd, 1 H, J = 4.6, J = 12.3 Hz), 4.26 (dd, 1 H, J = 4.8, J = 12.5 Hz), 4.13 (m, 2 H), 3.66 (2ddd, 2 H), 3.56 (s, 3 H), 3.06 (dd, 1 H, J = 8.7 Hz, J = 10.7 Hz), 2.08–1.99 (7 s, 21 H); $^{13}$C NMR (CDCl$_3$) δ 170.57–169.22 (CO), 103.79, 83.79, 76.03, 74.02, 73.11, 71.77, 71.48, 69.51, 68.25, 62.25, 62.15, 57.40, 49.90, 20.86–20.77 (CH3).

Third, methyl 2-S-β-D-glucopyranosyl-2-thio-β-D-glucopyranoside (methyl 2-thio-β-sophoroside; G2SG-OMe) (chemical structure in Supplementary Table 1): 1 M sodium methylate (200 μL) was added to a solution of methyl 2-thio-β-sophoroside heptaacetate (26 mg, 0.04 mmol) in MeOH (5 mL) and stirred at ambient temperature for 2 h. The mixture was neutralised with the Amberlite IR 120 H$^+$ resin, filtered and concentrated. After lyophilisation, pure methyl 2-thio-β-sophoroside was isolated with 97% yield (14.5 mg). [α]$_D$ −15.02 (c 0.3, H$_2$O), ESI-MS: m/z = 395 [M$^+$Na]$^+$; $^1$H NMR (CDCl$_3$) δ 4.76 (d, 1 H, J = 10.0 Hz), 4.57 (d, 1 H, J = 9.0 Hz), 3.96 (m, 2 H), 3.78 (m, 2 H), 3.66 (dd, 1 H, J = 8.8 Hz, J = 10.8 Hz), 3.61 (s, 3 H), 3.56–3.33 (m, 6 H), 2.85 (dd, 1 H, J = 9.2 Hz, J = 10.6 Hz); $^{13}$C NMR (CDCl$_3$) δ 102.36, 84.43, 79.76, 77.02, 75.61, 74.97, 72.82, 70.63, 69.32, 60.77 (2 C), 57.23, 51.26.

**Steady-state affinity binding with recombinant HvExoI.** Interaction between recombinant HvExoI and analytes Glc (molecular mass 180.2 g), G6SG-OMe (372.4 g) and PEG (n = 5–10); (380–420 g) were performed using Surface Plasmon Resonance (SPR) at 25 °C using the Biacore X100 V2.0.1 instrument (GE Healthcare, USA) equipped with the plus package. Purified recombinant HvExoI at 99% homogeneity (detected by SDS-PAGE) was covalently attached to a CM5 chip (GE Healthcare) in 10 mM sodium acetate buffer, pH 5.0. The carboxymethyl groups present on the dextran layer of the CM5 chip were activated with 1-ethyl-3-(3-dimethylaminopropyl) carbodiimide hydrochloride (EDC) and N-hydroxysuccinimide (NHS) (GE Healthcare). Any unoccupied reactive sites were blocked by a 7 min injection of 1 M ethanolamine hydrochloride-NaOH, pH 8.5. A ligand density of ~15.000 RU of recombinant HvExoI was achieved. Due to the fast rate constants of interactions, it was impossible to obtain individual rate constants $k_a$ and $k_d$. Hence equilibrium analyses were performed for all protein-analyte interactions. For each interaction serial dilutions of analytes covering the $10 \cdot K_D$–$0.1 \cdot K_D$ concentration ranges were prepared in 10 mM HEPES-NaOH buffer, pH 7.4 with 150 mM NaCl. Samples were injected onto the surface of a chip at a rate of 30 μl/min for 90 s, after which the formed enzyme/analyte complexes could dissociate for 90 s. As all analytes dissociated completely from the surface of the recombinant enzyme immobilised on a chip, no regeneration steps were needed. All responses were double referenced using a blank-immobilised reference surface (NHS + EDC-activated/deactivated) and a series of blank injections conducted to account for the system and injection artefacts. For each analyte at least one concentration was injected in duplicate. Triplicate experiments were performed, and the double-referenced data were globally fitted using a 1:1 equilibrium interaction model and the Biacore T200 v2.0 analysis software (GE Healthcare). Steady-state affinity binding constants $K_D$ (Supplementary Fig. 3) were calculated 30 s after the onset of injections using a 5 s window in an average response in the fit.

**Inhibition constants $K_i$ with native HvExoI.** $K_i$ values of deoxy-glucose and alkyl β-D-glucoside derivatives, and G2SG-OMe with native HvExoI were derived from hydrolytic rates. Enzyme inactivation was monitored at 30 °C by incubating 8 μmoles of the enzyme in 100 mM sodium acetate buffer, pH 5.25, containing 0.8–6.6 mM 4NPGlc as a substrate, 160 μg/ml BSA, and 3dGlc, 4dGlc, octyl-O-Glc, octyl-S-Glc and G2SG-OMe. Each inhibitor was tested at six concentrations at 0.4–3 times the $K_i$ values in duplicate. We used the $v = V_{max} S / [K_M \times (1 + I/K_i) + S]$ equation to calculate $K_i$ constant; $K_i$ is defined as a dissociation constant of the enzyme-inhibitor complex. The enzyme activity was monitored at 410 nm and Dixon plots were used to determine $K_i$ under the pseudo-first-order rate limiting conditions by a non-linear regression analysis, as described previously[22].

**Expression and crystallisation of recombinant HvExoI.** The optimised HvExoI cDNA (GenBank Accession GU441535; was subcloned into the pPICZαBNH$_8$ vector, from which the protein was expressed in Pichia pastoris strain SMD1168H (Invitrogen, Carlsbad, CA, USA). The R158A/E161A variant (primers listed in Supplementary Table 6) was prepared by site-directed mutagenesis and expressed in Pichia[24]. Recombinant HvExoI enzymes were purified using SP-Sepharose

cation-exchange and immobilised metal affinity chromatography[24,25]. Purified recombinant wild-type (WT) and variant HvExoI were crystallised via a hanging-drop vapour-diffusion method, using macro- and cross-seeding with WT native crystal seeds. The fully-grown crystals of recombinant HvExoI reached up to $500 \times 250 \times 375$ μm sizes after 5 to 14 days and were used for diffraction[25].

**Crystal structure determination.** Enzyme crystals with lengths of up to 400 μm in the longest dimensions were transferred in 100 mM HEPES-NaOH buffer, pH 7.0, containing 1.2% (w/v) PEG (n = 5–10) and 1.7 M ammonium sulphate (solution A), where 3-deoxy-glucose (3dGlc), 4-deoxy-glucose (4dGlc), n-octyl β-D-glucopyranoside (octyl-O-Glc), n-octyl 1-thio-β-D-glucopyranoside (octyl-S-Glc), methyl 2-thio-β-sophoroside (G2SG-OMe), methyl 6-thio-β-gentiobioside (G6SG-OMe) or Glc were supplied during ligand perfusions. The final concentrations of ligands were between 5–20 mM and perfusion proceeded during 5–720 min. Alternatively, the crystals were deposited in a solution A with the concentration of PEG increased to the 35% (v/v). After various perfusion times at $4 \pm 2$ °C crystals were cryo-protected with 15% (v/v) glycerol in the solution A and mounted on a synchrotron goniometer in a stream of N$_2$ gas at 100 K (Oxford Instruments, UK). X-ray diffraction data (native, and with 3dGlc and 4dGlc) were collected at the undulator beamline BioCARS 14-ID-B at the University of Chicago Advanced Photon Source (USA) equipped with a bent cylindrical Si-mirror (Rh coating) Diamond(111) double-bounce monochromator and focussing to the MARCCD-165 detector. X-ray diffraction data sets of crystals with PEG and with octyl-O-Glc and octyl-S-Glc were collected using the multipole wiggler beamline BL5 at the Photon Factory (Japan), which was fitted with a collimating mirror, double-crystal Si(111) monochromator and a focusing to the ADSC Quantum 315 CCD detector. X-ray diffraction data from recombinant WT and variants were collected at the MX1 and MX2 beamlines of the Australian Synchrotron (Australia) at 100 K (Oxford Instruments) or at ambient temperature (291 K) with a collimating mirror, double-crystal Si(111) monochromator 03BM1 dipole/bending magnet and ADSC Quantum 210r Detector. In the latter case, crystals were mounted using transparent polymer capillaries from the RT Screening kit (MiTeGen). All data were collected at 0.5–1° oscillations throughout 180–720°. Data were processed using the DENZO/SCALEPACK HKL 2000 suite of programmes[61]. Autoindexing determined that space group P4$_3$2$_1$2 of crystals were consistent with the tetragonal space group P43212 in all instances. The structures of the enzyme-ligand complexes were refined using CCP4 REFMAC5[62] and PHENIX[63]. We used the solved structures (PDB accessions: 1EX1, 1IEQ, 1IEV, 1IEW, 1IEX and 1J8V) without ligands, ions, glycerol and water molecules, as starting models for refinements of HvExoI in complex with ligands[7,15–17]. The iterative model building using XtalView/MIFIT[64], Coot[65] and refinements with REFMAC5[62] allowed tracing of all the residues. Following convergence in a standard refinement, a further improvement of about 2% in the Rwork/Rfree factors ratio was achieved by refining domain 1, (residues 1–357) and domain 2, (residues 374–559), as two independent anisotropic entities with translation-libration-screw (TLS) motion[66,67]. Electron densities for the ligands were well-defined in the active site regions at 3σ level in m|$F_o$|-|$F_c$| maps and water molecules were located automatically with CCP4 ARP at levels higher than 3σ. Water molecules were retained if they satisfied H-bond criteria and if their 2 m|$F_o$|-D|$F_c$| electron density maps were confirmed after refinements, where m is the figure of merit and D is an estimated coordinate error. During model building and refinement, 5% of the data were flagged for cross-validation to monitor the progress of refinements, using $R_{free}$ statistics[68,69]. The PROCHECK programme[69] was used to check the geometrical quality of the models. Ramachandran plots[70,71] of all structures showed that 99.8% of residues were found in the most favourable, additionally allowed and generously allowed regions of the plot, with well-defined electron density for Ile432, the only residue located in disallowed regions. Data collection, cell parameters and refinement statistics of all structures are summarised in Supplementary Tables 7–10. Graphics images were prepared with PyMol (Schrödinger, USA); electrostatic potentials were generated with a solvent probe radius at 1.40 Å, and protein and solvent dielectric constants 2 and 78, respectively.

**GC/MS with evaporative light scattering detection.** Six crystals of native HvExoI with well-defined edges in a longest axis dimension of ~80–120 μm were collected from a hanging drop in a cat whisker, deposited in a test tube, washed in solution A, pelleted by low speed centrifugation (500 × g, 1 min) and dissolved in 100 μl of 90% (v/v) ethanol. The mixture was treated for 5 min at 80–85 °C, held on ice for 1 h and centrifuged (4000 × g, 10 min). The supernatant was recovered, the pellet washed twice in 100 μl of re-distilled water (conductance 24 μS), and the supernatants were combined and freeze dried. The extract obtained from crystals, a freeze-dried mother liquor, and a mother liquor with authentic Glc (100 ng) being added, were dissolved in 25 μl 65% (v/v) CH$_3$CN and separated by high performance liquid chromatography (HPLC) coupled with the Evaporative Light Scattering Detector (ELSD model 800, Alltech Associates Inc.). The samples were eluted isocratically with 65% (v/v) CH$_3$CN at a flow rate of 0.6 ml/min. The fractions eluting between 6–7.3 min were collected and freeze dried. A model 1090 HPLC liquid system controlled by ChemStation software (Agilent Technologies) was used for chromatography, and separation was carried out on a Prevail

Carbohydrate ES column, 5 µm, 250 × 4.6 mm (Alltech Associates Inc.). The column temperature during separations was 21 °C. The column eluent was split in 6.5 (collect): 1 (detector) ratio. The detector drift tube was at 40 °C, the nitrogen inlet pressure was 1.5 bars and the signal GAIN setting was set at 4. The HPLC-eluted fractions (6–7.3 min) were reduced with 1 M NaBD$_4$ in 2 M NH$_4$OH for 18 h at 4 °C and acetylated with acetic anhydride. Myo-inositol (20 ng) was added as an internal standard. Acetylated alditols were analysed on a low polarity 25 m × 0.25 mm (internal diameter) Chrompack Capillary Column CP-Sil 5 CB LB/MS (Varian Inc., CA, USA) using He as a gas flow with a Hewlett-Packard 6890 Series GC System and a Hewlett-Packard 5973 mass selective detector (Agilent Technologies)[72].

**NMR spectroscopy of Glc bound to HvExoI.** NMR spectra were acquired at 283 K on the Bruker AVANCE III 600 MHz and 800 MHz spectrometers (Bruker, MA, USA). For Saturation Transfer Difference (STD) experiments, a stock solution of recombinant HvExoI deglycosylated by endoglycosidase H[25] in milliQ water (143 µM, 20 mM sodium acetate buffer, pH 5.25, 159 mM NaCl) was diluted to 40 µM in the same buffer in D$_2$O, and 60 equivalents (2.4 mM) of Glc from a stock solution in D$_2$O were added. For transferred Nuclear Overhauser Effect Spectroscopy (trNOESY) experiments, the same recombinant enzyme stock solution was diluted to 57 µM with a deuterated 20 mM sodium acetate buffer, pH 5.25, to which five equivalents of Glc (0.285 mM) were added followed by 300 msec mixing time. In a separate experiment, to the 57 µM recombinant enzyme stock solution, five equivalents of Glc and three equivalents of thiocellobiose (0.171 mM) were added, both from a stock solution in D$_2$O, followed by 300 msec mixing time. STD spectra were acquired with 2048 scans, using 30 ms Gaussian shaped pulses for selective protein saturation at 0.65 ppm; the reference spectrum employed off-resonant saturation at 100 ppm. Different saturation times (0.5, 1, 2, 3 and 5 s) were used. Prior to Free Induction Decay acquisition, an excitation sculpting double echo for water suppression and a T2 filter (100 ms XY16 with 500 µs echo times to minimise dephasing from JHH coupling evolution) protein signal suppression was inserted. STD spectra for free Glc were acquired under the same conditions to show that no direct ligand saturation occurred. trNOESY spectra were acquired with 64 scans, 4 K points in the direct dimension, 256 points in the indirect dimension and 300 ms mixing time. For water suppression we employed a WATERGATE scheme using 3–9–19 binomial pulses. $^1$H signal assignments of ligands were derived from two-dimensional TOtal Correlation SpectroscopY (TOCSY), NOESY and $^{13}$C-Heteronuclear Single Quantum Coherence (HSQC) spectra. The NOESY spectrum of free thiocellobiose was acquired at 313 K with 700 ms mixing time to obtain positive NOEs. A NOESY spectrum acquired under the conditions used for the trNOESY spectrum (800 MHz, 283 K, 300 ms mixing time) yielded no observable NOEs, indicating NOE zero-crossing for thiocellobiose (for $\tau c = 1.12/\omega 0$).

**Classical MD simulations.** The structure of native HvExoI in complex with Glc (defined as β-D-glucopyranose) (PDB 3WLH) with occupancy 0.5 for −1 and +1 subsites, suggests that the two subsites are alternatively occupied. We set up two enzyme-product complexes, with Glc bound in the −1 subsite, whereas the +1 subsite was occupied by water molecules, and vice versa. The protonation state of the titratable residues was selected based on their hydrogen bonding environment. Namely, all Arg, Lys, Asp and Glu residues were in ionised form, except for the acid-base residue Glu491, which was protonated (i.e. non-ionised) to represent the catalytic acid/base form required in the first step of the reaction, and the predominant form at the optimal pH of the enzyme. Histidine residues 98, 111, 234, 331, 419, 487 and 586 were considered as N$_\epsilon$-protonated and His207, His262 and His377 as N$_\delta$-protonated. Assigned titration states agreed with pK$_a$ values predicted by PROPKA3.1 and with the optimised hydrogen placement algorithm of MolProbity. Disulfide bridges were considered between Cys151 and Cys159, and between Cys513 and Cys518. Three sodium ions were included to achieve neutrality, in addition to 150 mM NaCl to mimic the physiological ionic strength. The system size consisted of approximately 150,000 atoms. The initial structures were equilibrated by classical MD simulations. Protein and Glc were described via the parm99SB[73] and the Glycam06[74] force fields, respectively, whereas the TIP3P model[70] was used for water molecules and Åqvist parameters for the sodium and chloride ions. The classical MD simulations were performed with NAMD2.9[76,77]. The SHAKE/RATTLE[78] algorithm was used to constrain the bonds involving hydrogen atoms in both the solvent and the solute. Multiple time step integration was carried out in r-RESPA; the production simulations used a base time step of 2 fs and a secondary time step of 4 fs for long-range interactions. Non-bonded interactions were computed at every step with a cut-off separation of 9 Å. Periodic boundary conditions and the particle mesh Ewald method were employed to evaluate long-range electrostatic interactions. Simulations were carried out in the NPT (N, number of particles; P, system pressure; T, temperature constant) ensemble, with a Nosé-Hoover thermostat and a Langevin piston Nosé-Hoover barostat[79,80] to maintain the temperature at 300 K and the pressure at 1 atm, respectively. Each of the two enzyme-product complex simulations was run for 40 ns without restraints, ensuring that the system reached equilibrium. Analyses of trajectories used VMD and in-house scripts. One representative snapshot from the last time window (≈5 ns) of each simulation was taken as the starting structure for further QM/MM MD simulations. Crystallographically observed $^4C_1$ Glc

conformations and protein-sugar interactions were consistently maintained during classical MD.

**QM/MM MD simulations.** Classical MD snapshots of two enzyme-product complexes were submitted to QM/MM MD simulations to re-equilibrate the system within the QM/MM Hamiltonian. The method by Laio and co-workers[81] was used, which combines Car-Parrinello MD[82] with force field-based MD. Only the atoms of Glc were included in the QM region (24 atoms), whereas the remainder of the protein and solvent were in the MM region. The MM system was treated with the Amber force field, for consistency with the previous classical MD simulations. The QM system was treated with Density Functional Theory (DFT), using a plane wave (PW) basis set with a kinetic energy cut-off of 70 Ry, Martins-Troullier pseudopotentials[83] and the Perdew-Burke-Ernzerhof (PBE) exchange-correlation functional[84]; this is at the same level theory as that described for sugar conformational Free Energy Landscapes (FELs)[33,34]. The QM system was enclosed in an isolated orthorhombic box of size 14.0 Å × 13.2 Å × 13.5 Å (−1 subsite) or 12.0 Å × 15.3 Å × 13.2 Å (+1 subsite). The electronic fictitious was set at 700 au and the simulation time step was at 0.12 fs. Both complexes were equilibrated at 300 K for around 5 ps before starting metadynamics simulations of Glc conformational FEL. A similar procedure was used in previous studies of conformational preferences of sugars in carbohydrate-active enzymes[34].

**Ab initio metadynamics simulations.** QM/MM metadynamics[85,86] simulations were performed to explore the conformational FEL of Glc bound in the −1 or the +1 subsites. We employed the metadynamics driver provided by the Plumed2 plugin, combined with the well-tempered metadynamics approach[87]. Ring-puckering coordinates qx and qy were used as collective variables, as described for isolated Glc[33]. Metadynamics parameters were selected from the analysis of evolution of the puckering coordinates during the unbiased QM/MM MD simulation and from expected energy barrier heights based on our previous work on both isolated Glc and a β-D-glucoside bound in the active site of 1,3;1,4-β-D-glucanase[34]. The height and the width of Gaussian terms were initially set at 1 kcal mol$^{-1}$ and 0.1 Å, respectively, and a bias factor of γ = 10 was used for the well-tempered approach. The deposition time was set at 24 fs (200 MD steps). The same set up was used for both enzyme-product complexes. Simulations were stopped after adding 7,668 Gaussians (enzyme-product complex with Glc bound in the −1 subsite) and 7725 Gaussians (Glc bound in +1 subsite). Analyses of evolution of puckering coordinates and FEL showed that the metadynamics simulation reached a diffusive regime and that the energy differences converged.

**Computational methods to explore the Glc exit path in HvExoI.** MD simulations in complex with Glc trapped in the −1 subsite (PDB 3WLH) proceeded as follows. Hydrogen atoms were added and protonation states of titrable residues at pH 7.0 were assigned by PROPKA3.1[88] through PDB2PQR version 2.0.0[89]. In this complex, Glu491 was modelled in a non-ionised form; this is the required protonation state for the acid/base catalyst in substrate/product complexes of retaining GH3 enzymes that catalyse hydrolysis by the generally accepted double-displacement reaction mechanism[22]. The entire system was solvated in a water box of 103.0 × 88.0 × 101.0 Å$^3$ volume, which ensured that there was a 5 Å thick layer of water molecules in each direction from any atom. Sodium ions were added to neutralise the total charge of the system forming a fully solvated model of 83,428 atoms. The system was energy-minimised to remove steric clashes and heated through 4 ns equilibration (integration steps of 2 fs) until reaching the temperature of 300 K at 1 Bar. This process also involved the application of positional restraints on the protein and ligand atoms that were gradually released until the whole system was fully equilibrated. Equilibration of the system was stopped when root-mean-square deviation (RMSD) values of the main chain reached the stable value lower than 1.60 Å. Two independent simulations of 100 ns each were run under NPT conditions, whereas temperature and pressure were controlled through Langevin dynamics and the Nosé-Hoover algorithm combined with the Langevin piston method[79,80], respectively. Periodic boundary conditions were applied, and the SHAKE algorithm[78] was used to adjust O-H separations of water molecules. The cut-off value of 12 Å was used for non-bonded interactions. All calculations were carried out with NAMD2.9[76,77] using the CHARMM22 force field[84,85] for the protein and TIP3P for water molecules[75], while the CHARMM36 all-atom carbohydrate force field[90] was used for Glc.

**Docking of Glc and disaccharides in HvExoI.** Molecular docking calculations of Glc and β-D-glucopyranosyl-(1,2)-D-glucose (G2OG) or β-D-glucopyranosyl-(1,3)-D-glucose (G3OG) to HvExoI for selected protein structures were performed with Gold5.2[91] applying a search cavity of the 20 Å radius and using the Goldscore scoring function. Side-chain flexibility for Glu491 was allowed, whereas Trp434 and Trp286 were flexible as specified below. Complex 1 was generated by docking Glc to the HvExoI:G2OG complex derived from the HvExoI:G2SG-OMe crystal structure (PDB 6MD6). Complex 2 was generated by docking G2OG to the HvExoI:Glc complex derived from the native HvExoI crystal structure (PDB 3WLH). Both complexes contain Glc bound at the −1 subsite and G2OG at +1 and putative +2 subsites. Complexes 3–6 were obtained by docking G2OG or G3OG to the HvExoI:Glc structure (PDB 3WLH) subjected to MD simulations,

with the major conformation of Trp434 (CA-CB-CG-CD1 dihedral angle of around −50°). MD simulations were carried out under periodic boundary conditions of NPT ensembles, as described for MD simulations of HvExoI in complex with Glc. Docking of G2OG, G3OG and β-D-glucopyranosyl-(1,6)-D-glucose (G6OG) lacking the Glc product in the active site was carried out as described above.

**GaudiMM calculations.** Ternary complexes 1–3 before MD simulation (as described above) and complex 3 after MD simulation with substrates bound in a productive mode, were used as starting points to investigate Glc displacement pathways from the −1 subsite of HvExoI. The protein contained Glc at the −1 subsite, and G2OG or G3OG attached at +1 and putative +2 subsites. GaudiMM[20] (https://github.com/insilichem/gaudi), a recently developed modular multi-objective genetic algorithm platform allows conformational exploration of defined genes (defined below) with multiple evaluation operators of fitness or objectives. The GPathFinder extension was used, defining as genes protein and substrates molecules, rotational bonds of ligands and rotamers of the protein, and as objectives, minimisation of van der Waals contacts and maximisation of separations between Glc geometric centres at the −1 subsite. This afforded the low-cost computational pre-identification of putative exit channels. The ProDy (Protein Dynamics)[92] Normal Mode Algorithm (NMA) was used to generate alternative starting structures for the GPathFinder. Normal mode calculations were performed with Tangram implementation on the UCSF Chimera suite[93], using residues grouped in clusters of 100 (to avoid protein distortion and achieve acceptable conformations), generating 10 modes with a cut-off of 15.0 Å and a Gamma LJ of 1.0. In the case of the crystal structure (PDB 3WLH), the second frame of the mode with frequency 31.2 cm$^{-1}$ was chosen, while the second frame of the mode with frequency 28.7 cm$^{-1}$ was selected for complex 3. In both cases, the displacement parameter in the Tangram interface was set to 150.

**PELE calculations.** The PELE software (https://pele.bsc.es)[21,94] was used to simulate Glc displacement and to identify the Glc migration route from the −1 subsite. PELE is a Monte Carlo based algorithm that generates configurations along a path through a sequential ligand and protein geometric perturbation scheme, side-chain conformational prediction and minimisation steps. The ligand (Glc) is perturbed by random translations and rotations, and by internal rotations of protein rotatable bonds, while Cα carbons of the protein are displaced following one of the six lowest Anisotropic Normal Modes[95]. The algorithm optimally arranges sidechains surrounding perturbed atoms using a rotameric library side-chain optimisation and energy minimisation with the OPLS-AA force field[96]. The move of the system is accepted, defining a minimum, or rejected according to a Metropolis criterion for a given temperature. The PELE scheme performs an effective exploration of the protein energy landscape[97–99]. Here, the exit path is defined by separations between the centre of masses of Glc molecules at the beginning (−1 subsite) and at the end of simulations (in solvent). Complexes 1–3 with Glc at the −1 subsite and G2OG or G3OG at +1 and putative +2 subsites were selected from MD simulations to identify potential exit pathways. One representative structure from each MD simulation of complexes 1–3 was selected as these complexes show G2OG or G3OG bound in the productive modes for subsequent catalysis. PELE simulations were terminated, when respective separations between centres of masses and C4-OH groups of Glc molecules in initial and final structures were above 15 Å.

**Reporting summary.** Further information on experimental design is available in the Nature Research Reporting Summary linked to this article.

## Data availability

The atomic coordinates and structure factors were deposited in the Protein Data Bank (www.pdb.org) with the PDB accessions: native HvExoI, and in complex with 3dGlc, 4dGlc, octyl-O-Glc, octyl-S-Glc and PEG400 are 3WLH, 3WLJ, 3WLK, 3WLM, 3WLN and 3WLL, respectively. The PDB accessions of WT recombinant ligand-free HvExoI and in complex with Glc, G2SG-OMe and G6SG-OMe are 3WLI, 3WLO, 6MD6 and 3WLP, respectively. The PDB accessions of recombinant R158A/E161A ligand-free HvExoI and in complex with Glc and G6SG-OMe, are 3WLQ, 3WLR and 6MI1, respectively. A reporting summary for this Article is available as a Supplementary Information file. All other data generated and/or analysed during the current study are available from the corresponding author upon reasonable request.

## Code availability

Software applications described in Methods with associated references were used without code modifications. For analyses of geometrical parameters along MD simulations, we have used an in-house script, which is available upon request from the corresponding author.

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

# ARTICLE

89. MacKerell, A. D. Jr., Feig, M. & Brooks, I. I. C. L. Extending the treatment of backbone energetics in protein force fields: limitations of gas-phase quantum mechanics in reproducing protein conformational distributions in molecular dynamics simulations. *J. Comput. Chem.* **25**, 1400–1415 (2004).

90. Guvench, O. et al. CHARMM additive all-atom force field for carbohydrate derivatives and their utility in polysaccharide and carbohydrate-protein modeling. *J. Chem. Theor. Comp* **7**, 3162–3180 (2011).

91. Jones, G., Willett, P., Glen, R. C., Leach, A. R. & Taylor, R. Development and validation of a genetic algorithm for flexible docking. *J. Mol. Biol.* **267**, 727–748 (1997).

92. Bakan, A., Meireles, L. M. & Bahar, I. ProDy: protein dynamics inferred from theory and experiments. *Bioinformatics* **27**, 1575–1577 (2011).

93. Pettersen, E. F. et al. UCSF Chimera—a visualization system for exploratory research and analysis. *J. Comput. Chem.* **25**, 1605–1612 (2004).

94. Madadkar-Sobhani, A. & Guallar, V. PELE web server: atomistic study of biomolecular systems at your fingertips. *Nucleic Acids Res.* **41**, W322–W328 (2013).

95. Cossins, B. P., Hosseini, A. & Guallar, V. Exploration of protein conformational change with PELE and metadynamics. *J. Chem. Theory. Comput.* **8**, 959–965 (2012).

96. Kaminski, G. A., Friesner, R. A., Tirado-Rives, J. & Jorgensen, W. L. Evaluation and reparametrization of the OPLS-AA force field for proteins *via* comparison with accurate quantum chemical calculations on peptides. *J. Phys. Chem. B* **105**, 6474–6487 (2001).

97. Hernández-Ortega, A. et al. Substrate diffusion and oxidation in GMC oxidoreductases: an experimental and computational study on fungal aryl-alcohol oxidase. *Biochem. J.* **436**, 341–350 (2011).

98. Lucas, F. et al. Molecular determinants for selective C25-hydroxylation of vitamins D2 and D3 by fungal peroxygenases. *Catal. Sci. Technol* **6**, 288–295 (2016).

99. Gygli, G., Lucas, M. F., Guallar, V. & van Berkel, W. J. H. The ins and outs of vanillyl alcohol oxidase: Identification of ligand migration paths. *PLoS Comput. Biol.* **13**, e1005787 (2017).

## Acknowledgements

We thank M. Raab (Slovak Academy of Sciences), B.J. Smith (La Trobe University) and G.B. Fincher (University of Adelaide) for interest in this research. H. Tong (Advanced Photon Source), N. Matsugaki and S. Wakatsuki (Photon Factory), and F. Pettolino (University of Melbourne) are thanked for advice and S. Pradeau for technical assistance. Funding was supported by the Huaiyin Normal University and the Australian Research Council (DP120100900) to M.Hrmova., the Australian Synchrotron (MX1 and MX2 beamlines) to M.Hrmova. and V.A.S., the Advanced Photon Source (14-ID-B beamline) to J.N.V. and M.Hrmova., the Photon Factory (BL5 beamline) to V.A.S, the Suranaree University of Technology and the Thailand Research Fund (BRG5980015) to J.R.K.C. and S.L., Generalitat de Catalunya (Commissioner for Universities and Research, Department of Innovation, Universities and Enterprise) and the European Union through a Beatriu de Pinós fellowship (BP-B 2013) to M.A.-P., Generalitat de Catalunya (SGR2017–1189), MICINN (CTQ2017–85496-P) and Spanish Structures of Excellence María de Maeztu (MDM-2017–0767) to C.R., Spanish Ministerio de Ciencia e Innovación to J.M.L. (CTQ2017–83745-P) and J.-D.M. (CTQ2017–87889-P), and Generalitat de Catalunya (SGR2017–13234) to J.-D.M. L.M. thanks to the Universitat Autònoma de Barcelona Talent Program, F.M. to CONICYT (PFCHA/Doctorado Becas CHILE/2012 72130118) and J.R.-G. to Generalitat de Catalunya and the European Social Fund (2017FI_B2_00168). S.F. was supported by Glyco@Alps (ANR-15-IDEX-02), Carnot Institut PolyNat, Labex ARCANE and CBH-EUR-GS (ANR-17-EURE-0003), and thanks to Chimie Moléculaire de Grenoble for access to facilities. M.A.-P. and C.R. acknowledge MareNostrum and Minotauro and BSC-CNS (RES-QCM-2017–2–001) resources. Use of synchrotrons was supported by the Australian Synchrotron Research Program, which is funded by the Commonwealth of Australia under the Major National Research Facilities Program.

## Author contributions

S.F. synthesised thio-oligosaccharides; M.Hrmova. performed enzyme inhibition kinetics; M.Hijnen. conducted SPR; S.L. and J.R.K.C. carried out mutagenesis; S.L. and M.H. performed crystallisation; V.A.S., J.N.V. and M.Hrmova. collected X-ray data; V.A.S., S.L., A.P. and M.Hrmova. refined crystal structures; M.Hrmova. executed GC/MS and HPLC; A.A. and J.J-B. performed NMR spectroscopy; I.T. conducted initial MD calculations; M.A.-P. and C.R. performed ab initio QM/MM metadynamics and conformational FEL calculations; F.M., L.T-S., J.-E.S.-A., J.R-G., J.M.L., J.-D.M. and L.M. implemented docking, classical MD simulations, GaudiMM and PELE calculations; M.Hrmova. wrote the manuscript with contributions from co-authors; M.Hrmova. oversaw the project.
