## [Peer Review File · Nature Communications]

Reviewers' comments:

Reviewer #1 (Remarks to the Author):

The study by Hrmova et al. provides fundamental insights into the mechanism of a plant glycoside hydrolase (family 3 glycoside hydrolase HvExoI). Glycoside hydrolases (GH) are involved in an enormous variety of biological processes, and understanding underlying mechanisms at the molecular level is important for making use of such enzymes for biotechnological processes, or to develop drugs modulating glycoside hydrolase activity. Using a combination of X-ray crystallography, QM/MM and MD simulations, NMR spectroscopy as well as other biophysical techniques Hrmova et al. succeed in developing a picture that shows how the exohydrolase HvExoI processes oligo- and polysaccharides, making use of the plasticity of the enzyme as well as the glycan substrate. The methods are sound, the results are novel, and the topic is of interest to a broad readership. Therefore, I recommend publication of this study in Nature Communications.

There are issues that should be addressed before publication.

General points

1/ The introduction needs to allude to at least some biological or biotechnological background of GHs. Two or three sentences would be sufficient. In this respect it would be interesting to learn how the authors extrapolate their findings to other GHs of the same or other families, which should go into the discussion or conclusions, of course. I don't think that it is helpful to speak of enzymes as of "nano-molecular devices that use protein architecture". For me, "nano-molecular devices" are artificial constructs or molecules, that e.g. make use of proteins as blueprints.

2/ It would greatly facilitate following the flow of thoughts if Fig. 1b and Fig. 1c would be cast into a separate figure aligned with the discussion of the NMR results.

3/ Very likely only a rather small fraction of the audience is familiar with conformational properties of carbohydrates or glycosidic linkages and with corresponding nomenclature. Therefore, I strongly recommend to include conformational formulas or simple modelling to explain the NOEs and trNOEs that nicely reflect conformational flexibility of carbohydrate substrates in the free state and freezing out bound conformations. This could be combined with supplemental Fig. S4.

4/ The use of enzyme isolated from seedlings for crystallography is of importance for some of the arguments leading to the model where the -1 Glc residue leaves the active site via a transient "channel". To my taste, this point should be explained in some more detail already in the results section.

Special points

1/ p. 2, "Notably, no other native ... to bind." I don't understand really why one couldn't use recombinant enzyme, soak it with glucose and come to similar conclusions.

2/ p.3, "Here, high-resolution ... go on to reveal ? ...". This sentence sounds somewhat "convoluted". Can it be split and transformed into something "easier-to-digest"?

3/ Fig. 1a: Dotted lines are not dotted in my copy. What is an "arrowed line"?

4/ Fig. 1b: It is very helpful to report saturation times and on/off resonance frequencies used for the STD NMR experiments. Same applies to mixing times used for the NOE/trNOE experiments.

5/ There are no references to STD and trNOE experiments in the methods sections. Unlike standard pulse sequences such as COSY or HSQC that are used for assigning NMR spectra these experiments are not really well known to a broader scientific community. Therefore, it is essential to at least quote the corresponding original work (STD: Mayer M, Meyer B. Characterization of Ligand Binding by Saturation Transfer Difference NMR Spectroscopy. *Angew Chem Int Ed* 38, 1784-1787 (1999); trNOE: Clore GM, Gronenborn AM. Theory and applications of the transferred nuclear overhauser effect to the study of the conformations of small ligands bound to proteins. *Journal of Magnetic Resonance* (1969) 48, 402-417 (1982).

6/ SPR data: As two Glc binding sites exist, wouldn't it make sense to speak of apparent dissociation constants? Does fitting improve if one would fit models to the data that take into account two or more binding sites?

7/ Only beta-Glc binds (Fig. 1b). Does that give a hint to a retaining mechanism?

Reviewer #2 (Remarks to the Author):

The work is an extension of their previous work on a beta-glucan hydrolase. Experimental 3D structural analysis and theoretical calculations were performed to understand the reaction of this enzyme. I felt that the manuscript is difficult to read. Each result is fragmented, and individual data are not supporting their conclusion in a well-organized manner. The glucose binding to this enzyme is not a new finding. This has been already reported in their previous work, although some results on the binding are added on this manuscript. Their idea of establishing the role bound glucose on the catalysis seems interesting, however the critical evidence is lacking to support their idea. Overall, I felt that the novelty of this paper is limited and not so strong to attract a broad audience. More specialized journal seems suitable, which is oriented to structural biology and enzyme catalysis.

Reviewer #3 (Remarks to the Author):

The manuscript describes a detailed and impressive experimental and computational study on a remarkable substrate/catalysis/product release process. It is well written and presented. In addition, to its multi-faceted approach, it also sparks discussion on multiple aspects, in addition to catalysis, such as enzyme evolution. However, there are a number of questions and comments that arise.

The authors state in the text and video that the enzyme modifies its structure *after* binding the second substrate. However, does this instead occur during binding of the second substrate? That is, is it a step-wise or concerted process?

Page 4 and 5: The authors state that the 4C1 conformation is low energy. However, is this the preferred conformation of the product in the less polar enzyme environment? If not, could any such conformational strain within the product contribute to the driving force for the release pathway and its development/progression?

Page 5 (bottom of page): Hydrogen bond lengths of 2.5 to 2.7 Å are not normally considered as indicative of tight binding. The authors describe the hydrogen bonds as "low-barrier", implying they are catalytic or facilitate highly fluxional proton transfer between the groups. But, would one expect to see transfer between the C6-OH and C4-OH groups and Aspartyl side-chain carboxylate given the difference in their pKa's?

When the authors state they attempted to model other conformations of the sugars but achieved convergence only with two, are they referring to the computational modelling or X-ray structure determination? If the former, was the forcefield tested to determine its ability to reliably map conformational preference in sugar rings?

Page 15: the authors state that the number of hydrogen bonds is retained as the product migrates. However, this does not necessarily mean that the strength of such bonds is retained along the pathway. Is there a gradient of hydrogen bonding strength or a gradient of polarity along the pathway? Or, is the lateral movement simply a physical process: a cavity opens and the product randomly walks out in the open solvent.

The authors (bottom of page 17) refer in the discussion to the idea that some enzymes may not have evolved an ability to exploit conformational changes while others, such as that studied, have not. Is this suggesting that the enzyme is ancient and if so can this be determined through evolutionary analysis? Or, does the process represent an alternate evolved process wherein enzymes use their own product to regulate enzyme specificity and turnover? The wording in the text is ambiguous and seems open to interpretation. In either case, this also raises questions as to what does the enzyme do in the very first case after it is synthesized; does it bind a variety of substrate(s) or mimics or does it bind its native substrate with higher specificity? Could this be modelled via Docking; i.e., docking substrate without bound product and comparing binding energy to when product is also bound?

The authors suggest, the product enhances substrate binding and specificity and conversely the substrate enables release of the product. Thus, instead of the phrase "substrate-assisted processive catalysis" (SAPC), is it in fact more correct to say "substrate-product (co)operative catalysis" (SPOC), or "substrate-product assisted catalysis" (SPAC)? The latter also connects with the well-established concept of substrate-assisted catalysis (SAC); expanding that concept to the idea that the product could also itself be involved in the process (thus the acronym is also functional).

SI, page 6, "compelx" should be corrected. What is the predicted pKa value of Glu491? Perhaps it should be noted in the text given that it is very uncommon (Note: it seems more correct to say it was made neutral as carboxylic acids are neutral and not normally referred to as protonated but rather non-ionized).

References 1 – 3, while they can be described as excellent and classics, seem a bit old; ideas about enzymes and their functioning have evolved since those times (see, for example, the work of A. Warshel and others).

The authors state that they took a snapshot from the end of each simulation (page 4 of methods). Given the nature of MD simulations, it is not necessarily always true that the final structure of an MD is a representative structure of the most common conformation. Was this checked and confirmed; if so it should be clearly and simply stated. They also note that the crystallographically observed product conformations and protein-sugar interactions were maintained ... does this mean they were constrained to be such or they were observed to be consistent over the course of the MD? Was more than one simulation of each complex run? Recent publications have suggested multiple runs may be preferred. A cut-off of 9 Å for non-bonded interactions has been suggested to be a bit short for modern simulations. Although, other articles have suggested it may suffice. Perhaps it was used due to being the default setting? 5 ps seems a relatively short time to equilibrate a system (see end of QM/MM-MD description). Has this been previously used successfully? If so, a citation might suffice to provide support to these methodology choices.

Page 5 of the methods the word "previosly" must be corrected.

Reviewers' comments:

The authors wish to thank the Reviewers for their constructive comments and suggestions that have helped us to improve the quality of the manuscript significantly.

Reviewer #1 (Remarks to the Author):

The study by Hrmova et al. provides fundamental insights into the mechanism of a plant glycoside hydrolase (family 3 glycoside hydrolase HvExoI). Glycoside hydrolases (GH) are involved in an enormous variety of biological processes, and understanding underlying mechanisms at the molecular level is important for making use of such enzymes for biotechnological processes, or to develop drugs modulating glycoside hydrolase activity. Using a combination of X-ray crystallography, QM/MM and MD simulations, NMR spectroscopy as well as other biophysical techniques Hrmova et al. succeed in developing a picture that shows how the exohydrolase HvExoI processes oligo- and polysaccharides, making use of the plasticity of the enzyme as well as the glycan substrate. The methods are sound, the results are novel, and the topic is of interest to a broad readership. Therefore, I recommend publication of this study in Nature Communications.

There are issues that should be addressed before publication.

General points

1/ The introduction needs to allude to at least some biological or biotechnological background of GHs. Two or three sentences would be sufficient.

Response: Several sentences on biological or biotechnological applications of glycoside hydrolases have been included in the Introduction section (page 2), where we re-structured the entire section (pages 2-3).

In this respect it would be interesting to learn how the authors extrapolate their findings to other GHs of the same or other families, which should go into the discussion or conclusions, of course.

Response: Firstly, we could only find one experimental evidence for the product entrapment in the native α -L-rhamnosidase classified in GH78 (Pachl et al. Crystal structure of native α -L-rhamnosidase from *Aspergillus terreus*. *Acta Crystallogr. D* 74, 1078-1084 (2018), although the authors could not explain the significance of their observation (page 3 of the Introduction section).

Secondly, in the Discussing section, we have explained why product or other co-factor entrapments have not been seen with other GH enzymes, and why the native structures carry that additional information could be beneficial for the understanding of catalytic cycles (page 11 of the Discussion section). This section reads: "The obvious explanation for not seeing naturally bound products or co-factors in structures of other GH enzymes is that these proteins are generated in recombinant hosts, where intracellular concentrations of potential enzyme reactants is not high-enough during protein maturation. To this end, solving crystal structures of enzymes purified from native sources offers an additional information that could be beneficial for the understanding of catalytic cycles."

Thirdly, we contemplated that "While this work is focussed on a plant GH3 hydrolase, we suggest that substrate-product assisted processive catalysis may be prevalent among exo-hydrolases with pocket-shaped active sites, irrespective of their substrate specificity." (page 14 of the Discussion section).

I don't think that it is helpful to speak of enzymes as of "nano-molecular devices that use protein architecture". For me, "nano-molecular devices" are artificial constructs or molecules, that e.g. make use of proteins as blueprints.

Response: The term "nano-molecular devices" has been removed and replaced by "biological catalysts" (page 2).

2/ It would greatly facilitate following the flow of thoughts if Fig. 1b and Fig. 1c would be cast into a separate figures aligned with the discussion of the NMR results.

Response: The data, originally included in Fig. 1 (panels **a**, **b** and **c**) have been casted independently, and are now uniquely presented in Fig. 2 (NMR data: Recombinant HvExoI recognises β -D-Glc in the 4C_1 conformation) and Fig. 5 (computational data: Conformational FEL maps of β -D-Glc bound in the active site of HvExoI).

3/ Very likely only a rather small fraction of the audience is familiar with conformational properties of carbohydrates or glycosidic linkages and with corresponding nomenclature. Therefore, I strongly recommend to include conformational formulas or simple modelling to explain the NOEs and trNOEs that nicely reflect conformational flexibility of carbohydrate substrates in the free state and freezing out bound conformations. This could be combined with supplemental Fig. S4.

Response: As suggested, we have extended Supplementary Fig. 4, in which we included the chemical formula of thiocellobiose in the chair representation (4C_1 conformation). Here, we have also illustrated short interproton distances in the Glc residue that produce NOE and trNOE effects; this material is presented in panel **d** of Supplementary Fig. 4. We have also shown schematically a pyranose ring in the 4C_1 conformation and indicated co-planar atoms. The addition to the legend reads: **d**, Left: short interproton distances in the Glc residue of thiocellobiose (arrows), which adopts the 4C_1 conformation produce NOE and trNOE effects. Right: a pyranose ring with the co-planar C2-C3-C5-O atoms in the 4C_1 conformation. (page 10 of Supplementary Information).

4/ The use of enzyme isolated from seedlings for crystallography is of importance for some of the arguments leading to the model where the -1 Glc residue leaves the active site via a transient "channel". To my taste, this point should be explained in some more detail already in the results section.

Response: In the Results section we have re-checked all sentences that relate to this point and included additional details (pages 8-10).

Special points

1/ p. 2, "Notably, no other native ... to bind." I don't understand really why one couldn't use recombinant enzyme, soak it with glucose and come to similar conclusions.

Response: This is an excellent point that allows us to explain that the motivation and the continuity of the research narrative is robust. The key observation that has led us to explore the product displacement route in HvExoI (and ultimately has led us to the discovery of the processive catalysis by HvExoI), was our original observation that the Glc product remains entrapped in the crystal structure of the native enzyme (Varghese et al., 1999; Hrmova et al. 2001 and 2002). We projected that the last remaining Glc from oligo- and polysaccharide substrate stays associated with the enzyme in a plant tissue, and only dissociates after an incoming substrate makes a contact with the surface of the enzyme.

Since we made this observation, we could not explain and describe the molecular mechanism for Glc displacement, up until recently, when advanced computational tools became available. We have extended the sentence (page 3) to reflect on the clarification of this point, and why this observation was absolutely critical for the current study.

We have also added two new sentences to emphasise the importance of the key finding that the Glc product is naturally bound to the enzyme in plant tissues (page 4).

2/ p.3, "Here, high-resolution ... go on to reveal ? ...". This sentence sounds somewhat "convoluted". Can it be split and transformed into something "easier-to-digest"?

Response: This sentence has been re-formulated, and its syntax simplified (page 3).

3/ Fig. 1a: Dotted lines are not dotted in my copy.

Response: We apologise. The line of the bottom trace was converted incorrectly during the generation of the pdf file and this line became a full line. We now provide the correct graphical representation with the bottom trace visualised in dots (Fig. 1a; top panel) (pages 4 and 28).

What is an "arrowed line"?

Response: Thank you for correcting the improper technical description of the line with arrows at each end. We have now used the appropriate terminology "left right arrow", which has been assigned to this type of arrows (Legend to Fig. 1; page 28).

4/ Fig. 1b: It is very helpful to report saturation times and on/off resonance frequencies used for the STD NMR experiments. Same applies to mixing times used for the NOE/trNOE experiments.

Response: We have provided the information on three parameters as requested. This information is contained in legend to Fig. 4 (page 30).

5/ There are no references to STD and trNOE experiments in the methods sections. Unlike standard pulse sequences such as COSY or HSQC that are used for assigning NMR spectra these experiments are not really well known to a broader scientific community. Therefore, it is essential to at least quote the corresponding original work (STD: Mayer M, Meyer B. Characterization of Ligand Binding by Saturation Transfer Difference NMR Spectroscopy. *Angew Chem Int Ed* 38, 1784-1787 (1999); trNOE: Clore GM, Gronenborn AM. Theory and applications of the transferred nuclear overhauser effect to the study of the conformations of small ligands bound to proteins. *Journal of Magnetic Resonance* (1969) 48, 402-417 (1982).

Response: These two fundamental references (References 30 and 31) have been included in the text of the manuscript (page 5).

6/ SPR data: As two Glc binding sites exist, wouldn't it make sense to speak of apparent dissociation constants?

Response: We are reporting the steady-state affinity K_D values that are presented in Table 1. Here, we have performed a steady state affinity analysis, that is calculated binding constants based on equilibrium measurements. We have now unified the terminology of steady-state affinity K_D values that we use in the text of the manuscript (page 5) and in the legend to Table 1 (page 34).

Does fitting improve if one would fit models to the data that take into account two or more binding sites?

Response: Yes, we agree. Most of the time the fit will improve when fitting a more complex fitting model as the model will have more degrees of freedom and thus will fit the data better. However, in our case this won't be appropriate as we fitted the simple equilibrium model, which is appropriate for our data. Fitting a more complex model, for example the two-site equilibrium model, is only appropriate if the two binding sites have equilibrium constants that differ at least by ten orders of magnitude (so one is fast and the other is slow), otherwise the fitted equilibrium constant won't accurately represent the interaction. In our case the best fit for our data was the single site interaction model.

7/ Only beta-Glc binds (Fig. 1b). Does that give a hint to a retaining mechanism?

Response: These data agree with our structural observations, where we could never refine α -D-Glc in the active site pocket, due to steric violations. These observations also agree with our earlier work published in *JBC*, where we used ^1H NMR to determine anomeric configuration of hydrolysis products; here we have revealed the retaining mechanism of HvExol; this also applies to an entire GH3 family of glycoside hydrolases (Reference 23). In this context we have added a short statement on the hydrolytic mechanism of HvExol (page 5).

Reference 23: Hrmova, M., Harvey, A. J., Wang, J., Shirley, N. J., Jones G. P. et al. Barley β -D-glucan exohydrolases with β -D-glucosidase activity. Purification and determination of primary structure from a cDNA clone. *J. Biol. Chem.* 271, 5277-5286 (1996).

Reviewer #2 (Remarks to the Author):

The work is an extension of their previous work on a beta-glucan hydrolase. Experimental 3D structural analysis and theoretical calculations were performed to understand the reaction of this enzyme. I felt that the manuscript is difficult to read. Each result is fragmented, and individual data are not supporting their conclusion in a well-organized manner. The glucose binding to this enzyme is not a new finding. This has been already

reported in their previous work, although some results on the binding are added on this manuscript. Their idea of establishing the role bound glucose on the catalysis seems interesting, however the critical evidence is lacking to support their idea. Overall, I felt that the novelty of this paper is limited and not so strong to attract a broad audience. More specialized journal seems suitable, which is oriented to structural biology and enzyme catalysis. **Response:** *We have not found here any specific comments or suggestions from Reviewer 2 that we could address. However, we have reviewed the manuscript and revised the connectivity of the data in individual sections, such that the manuscript narrative is more fluent.*

Reviewer #3 (Remarks to the Author):

The manuscript describes a detailed and impressive experimental and computational study on a remarkable substrate/catalysis/product release process. It is well written and presented. In addition, to its multi-faceted approach, it also sparks discussion on multiple aspects, in addition to catalysis, such as enzyme evolution. However, there are a number of questions and comments that arise.

The authors state in the text and video that the enzyme modifies its structure *after* binding the second substrate. However, does this instead occur during binding of the second substrate? That is, is it a step-wise or concerted process?

Response: *In this specific case, we have defined the conformational change of Trp434, compared to its conformation in crystal structures (for example 3WLH – the native complex with Glc). However, because this change was already observed in simulations before an incoming substrate binds, this rotation may be indicative of a conformational selection process in a step-wise manner. The accompanying text in Supplementary Video 1 has been modified accordingly.*

The change in the Tyr253 sidechain orientation was observed in simulations of complexes with the incoming substrate bound, but not when only Glc was present. In those simulations (e.g. Complex 3; page 7 of the Supplementary Information document) that started with the G2OG or G3OG substrates bound at the +1 and +2 subsites, the rotation of Tyr253 sidechain occurred few ns after binding of substrates at these +1 and +2 subsites. This suggested, as mentioned in the text of the manuscript (The Results section, page 8), that "... binding of the G2OG or G3OG substrates triggered the conformational change of Tyr253...". This rotation of Tyr253 sidechain enlarges the lateral cavity adjacent to the -1 subsite where Glc will migrate, thus creating a suitable exit route. These structural changes that occur in a stepwise manner, affect the toll-like barrier formed by Glu491, Asp285 and Arg158. Respective changes have been made in the manuscript (pages 8-10 of the Results section, and pages 12-13 of the Discussion section) and in Supplementary Video 1, to better describe these structural changes.

Page 4 and 5: The authors state that the 4C1 conformation is low energy. However, is this the preferred conformation of the product in the less polar enzyme environment? If not, could any such conformational strain within the product contribute to the driving force for the release pathway and its development/progression?

Response: *Yes, the conformational FEL MD calculations of Glc bound in the enzyme's active site showed that the ⁴C₁ conformation was the lowest energy one (by 8 kcal/mol more stable than the ¹S₃/B_{3,0} conformation and with a free energy barrier for conversion of 11 kcal/mol) (page 6 and Fig. 5, left and middle panels).*

Further, the QM/MM simulations were performed in the enzyme environment, and thus we considered the polarity of the enzyme and the specific interactions formed by the Glc molecule in each subsite. Indeed, panels A and B in Fig. 5, corresponding to FEL of Glc in the -1 and +1 subsites show that the enzyme restricts the number of available conformations for the sugar, compared to that of isolated Glc (Fig. 5C, adapted from Reference 33).

*Reference 33: Biarnés, X., Ardèvol, A., Iglesias-Fernández, J., Planas, A. & Rovira, C. Catalytic itinerary in 1,3-1,4-β-glucanase unraveled by QM/MM metadynamics. Charge is not yet fully developed at the oxocarbenium ion-like transition state. J. Am. Chem. Soc. **133**, 20301-20309(2011).*

The QM/MM simulations also show that the 4C_1 conformation is the low energy conformation in the -1 and +1 subsites, as shown in Fig. 5A and 5B, respectively. However, the relative populations of other (higher energy) distorted conformations vary, and thus we cannot exclude that the conformational strain caused by the changes in the conformer distribution could aid in driving the Glc product away from the active site, as pointed out by the Reviewer. Nonetheless, the relative population of conformers changes only slightly as the sugar moves from one subsite to the other as shown Fig. 5 (panels A and B), indicating that other factors, e.g. disruption and/or establishment of hydrogen bonds and stacking interactions, may play a role in the product release. In other words, both the energy released upon conformational changes and upon protein-sugar interactions would contribute to the force that drives the sugar away from the active site.

Page 5 (bottom of page): Hydrogen bond lengths of 2.5 to 2.7 Å are not normally considered as indicative of tight binding. The authors describe the hydrogen bonds as "low-barrier", implying they are catalytic or facilitate highly fluxional proton transfer between the groups. But, would one expect to see transfer between the C6-OH and C4-OH groups and Aspartyl side-chain carboxylate given the difference in their pKa's?

Response: Thank you for this comment. We accept the Reviewer's critique that the hydrogen bond lengths of 2.5-2.7 Å are too long to be considered as indicative of tight binding. We are also aware of the danger of using the term "low-barrier hydrogen bonds" in the context of our discussion. We have removed this term (and Reference) from the manuscript. Instead this term, we are using the term "short H-bonds" (pages 5 and 7).

When the authors state they attempted to model other conformations of the sugars but achieved convergence only with two, are they referring to the computational modelling or X-ray structure determination? If the former, was the forcefield tested to determine its ability to reliably map conformational preference in sugar rings?

Response: In this section we are referring to the crystallographic refinements through CCP4, using our measured X-ray experimental data. In other words, we have conducted refinements of β -D-glucose molecules in the -1 and +1 subsites, through the positioning of a variety of conformers (and their combinations) into the electron density map. After individual refinements, a residual density was present when using the 3S_5 , 1S_5 , $B_{3,0}$ conformers (and their combinations). On the other hand, after the positioning the 4C_1 (occupancy 0.8) and ${}^1S_3/B_{3,0}$ (occupancy 0.2) conformers in the electron density map, the structural refinements reached convergence. These refinements indicated that the β -D-glucose molecules contained alternate 4C_1 chair and 1S_3 skew-boat conformers, and most likely these two conformers are in a dynamic equilibrium. This point was clarified and explained, as suggested by the Reviewer on page 5.

It is also worthy to comment that the conformational FEL maps presented in Fig. 5A and 5B (referring to the -1 and +1 subsites) support the crystallographic modelling based on the electron density. These conformations are the only ones that are accessible in the restricted environment of the enzyme, with the 4C_1 conformation as the one with the lowest in energy (and thus higher occupancy) and the ${}^1S_3/B_{3,0}$ distorted conformation higher in energy (i.e. lower occupancy).

We have also cross-checked the entire manuscript and clearly indicated, where we are referring to convergence in refinements of experimental data and in computational modelling.

Page 15: the authors state that the number of hydrogen bonds is retained as the product migrates. However, this does not necessarily mean that the strength of such bonds is retained along the pathway. Is there a gradient of hydrogen bonding strength or a gradient of polarity along the pathway? Or, is the lateral movement simply a physical process: a cavity opens and the product randomly walks out in the open solvent.

Response: We apologise for the misleading statement. The point is that the displacement pathway includes polar residues, which interact with Glc through H-bonds and that the presence of such residues in the active site and the lateral cavity seem to be a conserved feature in GH3 enzymes (Supplementary Figure 9). We have performed the detailed analysis of H-bonds formed between the atoms of Glc and protein residues and the atoms of Glc and G2OG/G3OG substrates at representative points along the displacement pathway. These new data

(Supplementary Tables 1-3) indicate that the numbers of H-bonds are not conserved. We have now removed the misleading statement from the manuscript (page 10 of the Results section).

The authors (bottom of page 17) refer in the discussion to the idea that some enzymes may not have evolved an ability to exploit conformational changes while others, such as that studied, have not. Is this suggesting that the enzyme is ancient and if so can this be determined through evolutionary analysis? Or, does the process represent an alternate evolved process wherein enzymes use their own product to regulate enzyme specificity and turnover? The wording in the text is ambiguous and seems open to interpretation.

Response: *Thank you for these constructive comments.*

The evolution of the existence of the substrate-product assisted processive catalytic mechanism in the GH3 family of hydrolases, currently with nearly 23,000 entries, requires a thorough and independent study. We project that this analysis should be extended to all GH enzymes that process oligo- and polysaccharides, and have a pocket-shaped catalytic site. We think that this analysis would require a significant bioinformatics effort and could represent the future direction of research for GH3 and other classes of enzymes. Now, that we have suitable tools, as presented in our work, we believe that these studies could be accomplished in near future. We do like to think that the described catalytic mechanism is ancient, however, this needs to be investigated through an exhaustive bioinformatics analysis. We have added the relevant statement to the Discussion section (page 12 of the Discussion section).

In Supplementary Fig. 9, we present the conservation of amino acid residues in HvExo1 classified in the GH3 family. We noted that the three residues that constitute the toll-like gate (Arg158, Asp285, Glu491; and Glu161 near Arg 158) that controls the passage of the Glc product from the -1 subsite to the lateral displacement cavity are absolutely conserved (Supplementary Fig. 9a), when examining 500 sequences with 35-95% sequence identity to HvExo1. It also seems that the key residues forming the internal walls of the lateral cavity (Supplementary Fig. 9b) are also conserved. For these reasons, it would appear that a similar catalytic mechanism may be more prevalent among exo-hydrolases with pocket-shaped active sites, irrespective of their substrate specificity (page 14 of the Discussion section).

In either case, this also raises questions as to what does the enzyme does in the very first case after it is synthesized; does it bind a variety of substrate(s) or mimics or does it bind its native substrate with higher specificity? Could this be modelled via Docking; i.e., docking substrate without bound product and comparing binding energy to when product is also bound?

Response: *We have solved several structures of recombinant HvExo1 (non-perfused active site contains water molecules) in complex with non-hydrolysable thio-analogues (two of them presented in the current work; PDB 6MI1 and 6MD6). These structures demonstrate that in the absence of the Glc product, the incoming substrate G6SG-OMe binds at the -1 and +1 subsites with tight binding affinity ($K_D=0.008 \times 10^{-3}$ M; Table 1; PDB 6MI1) binds at the -1 and +1 subsites, while the G2SG-OMe substrate with a much lower binding affinity ($K_i=2.55 \times 10^{-3}$ M; Table 1; PDB 6MD6) binds at the +1 and +2 subsites (due to a thio-glycosidic bond rigidity); please note that although we compare K_D and K_i values to illustrate the strength of binding, we are aware of conceptual differences between these parameters.*

Further, as suggested, we conducted further calculations to find out, if docking of substrates lacking the bound Glc product in the active site would lead to more-or-less efficient binding than that with Glc bound in the -1 subsite. We found out that docking of the disaccharide substrates G2OG, G3OG and β -D-glucopyranosyl-(1,6)-D-glucose (G6OG) to the -1 and +1 subsites predicted higher binding affinities (Goldscore scoring function values of 66 for G2OG, 76 for G3OG and 74 for G6OG), when the Glc product was absent in the active site. However, when the Glc product was included in the -1 subsite, G2OG, G3OG and G6OG at the +1 and putative +2 subsites formed weaker interactions (Goldscore scoring function values of 60 for G2OG, 57 for G3OG and 61 for G6OG, which are indicative of lower binding affinities). We concluded that bound Glc lowered binding energies for incoming substrates, as they had no access to the higher affinity -1 subsite (page 9). We also added a sentence on the methodology of docking in the Methods section (page 20).

The authors suggest, the product enhances substrate binding and specificity and conversely the substrate enables release of the product. Thus, instead of the phrase "substrate-assisted processive catalysis" (SAPC), is it in fact more correct to say "substrate-product (co)operative catalysis" (SPOC), or "substrate-product assisted catalysis" (SPAC)? The latter also connects with the well-established concept of substrate-assisted catalysis (SAC); expanding that concept to the idea that the product could also itself be involved in the process (thus the acronym is also functional).

Response: *We have critically re-considered the term "substrate-assisted processivity" that we coined for the newly discovered catalytic mechanism of the HvExoI enzyme with a pocket-shaped catalytic site. We agree with the Reviewer's suggestion that the substrate too must be considered, as both are involved concurrently in evocation and formation of a transient and autonomous lateral cavity, which serves as a conduit for the Glc product departure. For this reason, we have adopted a new term, as suggested by Reviewer: "substrate-product assisted processive catalysis", that we now use throughout the manuscript.*

SI, page 6, "compelx" should be corrected.

Response: *The error (complex) on page 6 of the Supplementary Information document has been corrected.*

What is the predicted pKa value of Glu491? Perhaps it should be noted in the text given that it is very uncommon (Note: it seems more correct to say it was made neutral as carboxylic acids are neutral and not normally referred to as protonated but rather non-ionized).

Response: *It is known that pKa values of the acid/base residues in GH enzymes are difficult to predict using standard computational tools. However, it is known from experiments in some GH enzymes that the acid/base residue is neutral (non-ionised), as it has a higher pKa constant value than that of the catalytic nucleophile:*

References:

- *Anderson, D. E., Lu, J., McIntosh, L. P., & Dahlquist, F. W. (1993) in NMR of Proteins (Clare, G. M., & Gronenborn, A. M., Eds.) pp 258-304, MacMillan Press, London.*
- *McCarter J. D. & Withers, S. G. (1994) Mechanisms of enzymatic glycoside hydrolysis. Curr. Opin. Struct. Biol. 4, 885-892.*
- *McIntosh L. P. et al. (1996) The pKa of the general acid/base carboxyl group of a glycosidase cycles during catalysis: A 13C-NMR study of Bacillus circulans xylanase. Biochemistry (USA) 35, 9958-9966.*
- *Rye, C. S. & Withers S. G. (2000) Glycosidase mechanisms. Curr. Opin. Chem. Biol. 4, 573-580.*

In fact, we have experimentally determined pKa constants for both catalytic acid residues (Reference 16), so the catalytic acid/base would present a significant population of the non-ionised form at pH=7.0, which protonation state is consistent with the catalytic mechanism of this retaining glycoside hydrolase.

Reference 16: Hrmova, M., DeGori, R., Smith, B. J., Vasella, A., Varghese, J. N. et al. Three-dimensional structure of the barley β -D-glucan glucohydrolase in complex with a transition state mimic. J. Biol. Chem. 279, 4970-4980 (2004).

Further, as stated above in the answer to Reviewer 1, we have also determined that the anomeric configuration of reaction products hydrolysed by HvExoI is retained, using $^1\text{H-NMR}$ (Reference 23).

Reference 23: Hrmova, M., Harvey, A. J., Wang, J., Shirley, N. J., Jones G. P. et al. Barley β -D-glucan exohydrolases with β -D-glucosidase activity. Purification and determination of primary structure from a cDNA clone. J. Biol. Chem. 271, 5277-5286 (1996).

In the light of the note above, we have included the sentence referring to the protonation state of Glu491 (page 19 of the Methods section). This text reads: "In this complex, Glu491 was modelled in a non-ionised form; this is the required protonation state for the acid/base catalyst in substrate/product complexes of retaining GH3 enzymes that catalyse hydrolysis by the generally accepted double-displacement reaction mechanism" (Reference 23).

References 1 – 3, while they can be described as excellent and classics, seem a bit old; ideas about enzymes and their functioning have evolved since those times (see, for example, the work of A. Warshel and others).

Response: Thank you very much for this comment. We have included two new references that reflect the increasingly accepted view on enzyme catalysis (page 2). To this end, we have cited two works of Professor Arieh Warshel:

Reference 1: Warshel, A., Sharma, P. K., Kato, M., Xiang, Y., Liu, H. & Olsson, M. H. Electrostatic basis for enzyme catalysis. Chem. Rev. 10, 3210-3235 (2006).

Reference 2: Adamczyk, A. J., Cao J., Kamerlin, S. C. & Warshel, A. Catalysis by dihydrofolate reductase and other enzymes arises from electrostatic preorganization, not conformational motions. Proc. Natl. Acad. Sci. USA 108, 14115-14120 (2011).

The authors state that they took a snapshot from the end of each simulation (page 4 of methods). Given the nature of MD simulations, it is not necessarily always true that the final structure of an MD is a representative structure of the most common conformation. Was this checked and confirmed; if so it should be clearly and simply stated.

Response: Reviewer is correct, and we have checked that the final MD structure is the representative of the conformations sampled during the production dynamics. For instance, the enzyme-sugar interactions present in this snapshot are consistently maintained during the production MD simulation run and agree with those present in the crystal structure. The text has been modified accordingly (page 17, the Methods section).

They also note that the crystallographically observed product conformations and protein-sugar interactions were maintained ... does this mean they were constrained to be such or they were observed to be consistent over the course of the MD?

Response: No restraints/constraints were applied during the production MD simulation run (40 ns), in which neither the product conformations nor the protein-sugar interactions deviated from the crystallographically observed parameters.

Was more than one simulation of each complex run? Recent publications have suggested multiple runs may be preferred.

Response: We did not perform more than one run of the complex, since our aim was not to exhaustively explore the complex dynamics, but to obtain an equilibrated structure from which to start the QM/MM MD simulations. A similar "one-replica" protocol has been successfully used in our previous works (reviewed in Reference 34). Moreover, given that the protein-sugar conformation was not observed to deviate significantly from the crystal structure and that the enzyme fluctuations at the active site were small, we think that this approach represents a good approximation.

Reference 34: Ardèvol, A. & Rovira, C. Reaction mechanisms in carbohydrate-active enzymes: Glycoside hydrolases and glycosyltransferases. Insights from ab initio quantum mechanics/molecular mechanics dynamic simulations. J. Am. Chem. Soc. 137, 7528-7547 (2015).

A cut-off of 9 Å for non-bonded interactions has been suggested to be a bit short for modern simulations. Although, other articles have suggested it may suffice. Perhaps it was used due to being the default setting?

Response: We agree with the Reviewer that modern computational approaches would allow to use larger cut-off values. More specifically, the default cut-off in Amber11 is 8 Å, but we increased slightly our cut-off to 9 Å following the recommendation of the NAMD manual for simulations with the Amber force field; please see this information:

<https://www.ks.uiuc.edu/Research/namd/2.9/ug/node13.html>; http://ambermd.org/namd/namd_amber.html.

Moreover, the cut-off of 9 Å was used for consistency with our previous studies, in which we also analysed the conformational free energy landscape (FEL) of substrates in GH enzymes (References 33 and 34).

Reference 33: Biarnés, X., Ardèvol, A., Iglesias-Fernández, J., Planas, A. & Rovira, C. Catalytic itinerary in 1,3-1,4-β-glucanase unraveled by QM/MM metadynamics. Charge is not yet fully developed at the oxocarbenium ion-like transition state. *J. Am. Chem. Soc.* **133**, 20301-20309 (2011).

Reference 34: Ardèvol, A. & Rovira, C. Reaction mechanisms in carbohydrate-active enzymes: Glycoside hydrolases and glycosyltransferases. Insights from *ab initio* quantum mechanics/molecular mechanics dynamic simulations. *J. Am. Chem. Soc.* **137**, 7528-7547 (2015).

5 ps seems a relatively short time to equilibrate a system (see end of QM/MM-MD description). Has this been previously used successfully? If so, a citation might suffice to provide support to these methodology choices.

Response: A similar time length (or even shorter time frames) were successfully used in our previous studies of the conformational FEL of sugars and the reactivity of GH enzymes (reviewed in Reference 34; please see above). Further, the system described here was already pre-equilibrated using classical, force field-based MD (~40 ns) and that the subsequent 5 ps QM/MM MD simulation was performed only to re-equilibrate the active site with the QM/MM Hamiltonian.

Additionally, we have specified the cut-off value for calculating of non-bonded interactions in MD simulations of the HvExo1:Glc and HvExo1:Glc:G2OG/G3OG complexes. This information has now been added to the Methods section (page 19). This new sentence reads: "The cut-off value of 12 Å was used for non-bonded interactions."

Page 5 of the methods the word "previosly" must be corrected.

Response: We apologise for this typo. However, we decided to remove the word 'previously', as it was not required, and instead used the word "described". Here we have used the Reference 33 (page 19 of the Methods section).

Reference 33: Biarnés, X., Ardèvol, A., Iglesias-Fernández, J., Planas, A. & Rovira, C. Catalytic itinerary in 1,3-1,4-β-glucanase unraveled by QM/MM metadynamics. Charge is not yet fully developed at the oxocarbenium ion-like transition state. *J. Am. Chem. Soc.* **133**, 20301-20309 (2011).

REVIEWERS' COMMENTS:

Reviewer #3 (Remarks to the Author):

The authors have addressed some of the concerns. Any further revisions are more minor than major (see below)

However, one concern that is alluded to or explicitly stated by more than one reviewer is with regards to novelty. This is still a challenge; is this the first case of SPAC and if not how explicitly does it go beyond what has previously been known or understood?

The broader picture is still unclear perhaps; how does this concept relate to the larger body of knowledge of biocatalysis?

Additional comments from Reviewer #3 (Remarks to the Author):

The authors have addressed some of the concerns. Any further revisions are more minor than major (see below). However, one concern that is alluded to or explicitly stated by more than one reviewer is with regards to novelty. This is still a challenge; is this the first case of SPAC and if not how explicitly does it go beyond what has previously been known or understood?

Response: To our knowledge, the substrate-product assisted processive catalytic mechanism has never been described in the literature with any enzyme. The reason for this is that typically products either diffuse rapidly from active sites, and thus are never trapped, or that these products in limited instances remained trapped and are observed in the active crystal structures sites (for example Reference 15), but these observations are never explained or reconciled. Further, observing the entrapped products in enzyme active sites requires the crystal structures resolved from native enzyme sources (where enzymes naturally catalyse reactions), which with the advent of recombinant technologies, happens only rarely.

*Reference 15: Pachi, P., Škerlová, J., Šimčíková, D., Kotik, M. & Křenková, A. et al. Crystal structure of native α -L-rhamnosidase from *Aspergillus terreus*. Acta Crystallogr. D74, 1078-1084 (2018).*

Additionally, crystal structures could capture entrapped products or reveal the dispositions of bound substrates in the active sites, but they could not capture intermediary states that occur between product entrapments and substrate binding events, as these processes occur rapidly. For these reasons, we suggest that our approach of multi-scale molecular modelling of nanoscale reactant movements, based on the high-resolution structures of product and substrate complexes, as presented in our work, is one of the advantageous strategies for unravelling these processes.

For the above-stated reasons, our discovery of the substrate-product assisted processive catalytic mechanism, as it is described in the current work, is a novel phenomenon. We have pointed to the novelty of our work in Abstract (page 2) and in the Introduction section on page 3.

The broader picture is still unclear perhaps; how does this concept relate to the larger body of knowledge of biocatalysis?

Response: We suggest that substrate-product assisted processive catalysis of enzymes with pocket-shaped catalytic sites will have much broader significance in catalysis of enzymes other than those of glycoside hydrolase classes, due to the plasticity of protein structures. It is exactly this plasticity that could be accountable for product dissociations, although in other instances, the precise atomic details of movements will be different, but the principle could hold. We have included these thoughts in a succinct form in the Discussion section on page 14.

Further, as for the evolutionary implications, we have extended our thoughts on this point in the Discussion section, where we pointed to the evolutionary origin of the newly discovered catalytic mechanism in plants. This sentence reads (page 14):" Preliminary analyses of 500 sequences related to HvExo1 revealed that in plants, this mechanism has evolved for the first time in land plants about 470 million years ago, while it is absent in cyanobacteria, and red and green algae."